# Homozygous frameshift mutations in *FAT1* cause a syndrome characterized by colobomatous-microphthalmia, ptosis, nephropathy and syndactyly

Najim Lahrouchi[1]

A failure in optic fissure fusion during development can lead to blinding malformations of the eye. Here, we report a syndrome characterized by facial dysmorphism, colobomatous microphthalmia, ptosis and syndactyly with or without nephropathy, associated with homozygous frameshift mutations in *FAT1*. We show that *Fat1* knockout mice and zebrafish embryos homozygous for truncating *fat1a* mutations exhibit completely penetrant coloboma, recapitulating the most consistent developmental defect observed in affected individuals. In human retinal pigment epithelium (RPE) cells, the primary site for the fusion of optic fissure margins, FAT1 is localized at earliest cell-cell junctions, consistent with a role in facilitating optic fissure fusion during vertebrate eye development. Our findings establish *FAT1* as a gene with pleiotropic effects in human, in that frameshift mutations cause a severe multi-system disorder whereas recessive missense mutations had been previously associated with isolated glomerulotubular nephropathy.

Correspondence and requests for materials should be addressed to C.R.B. (email: c.r.bezzina@amc.uva.nl) or to B.P.B. (email: brooksb@nei.nih.gov) or to A.S. (email: sefianigen@hotmail.com). #A full list of authors and their affiliations appears at the end of the paper.

The eye develops as an evagination of the neural plate, which subsequently invaginates to form a dual-layered optic cup. This invagination is asymmetric, and a ventral opening (optic fissure) forms around the 5th week of human gestation[1]. For the eye to develop normally, the two edges of the fissure must approximate and fuse. If the optic fissure fails to fuse, uveal coloboma, a potentially blinding congenital malformation, results. Uveal coloboma accounts for up to 10% of childhood blindness worldwide, affecting between 0.5 and 2.6 per 10,000 births[1]. Mutations in several developmentally regulated genes, including *CHD7, CHX10/VSX2, GDF3, GDF6, MAF, OTX2, PAX2, PAX6, RAX, SHH, SIX3, SOX2, FOXE3, STRA6, BCOR, BMP4, YAP1,* and *MITF,* have been reported in patients with uveal coloboma[2]. To our knowledge the *FAT1* gene has not been previously associated with microphthalmia and coloboma.

The *FAT* cadherins are involved in fundamental developmental processes including cell–cell contact[3], planar cell polarity[4], cell migration[5], and maintenance of apical–basal polarity[6] in epithelial cells. Loss of FAT1 function causes decreased epithelial cell adhesion and podocyte foot process effacement, resulting in abnormal glomerular filtration and nephropathy in humans and mouse, and cystic kidney in zebrafish[7,8]. *Fat1*$^{-/-}$ mice also display partially penetrant anterior neural tube closure defects due to reduced actin accumulation, leading to apical constriction defects in the neural epithelium[9]. These developmental observations in epithelial cell types suggest that *FAT1* plays an important role in epithelial cell–cell adhesion and/or sheet fusion. Epithelial sheet fusion is one of the most critical morphogenetic events occurring during embryonic development, failure of which causes clinically well-characterized congenital defects including, neural tube closure defects (e.g. spina bifida), secondary palatal epithelial fusion defects (e.g. cleft palate), defective fusion of bilateral urogenital primordia (e.g. hypospadias), and optic fissure closure defects (e.g. coloboma)[10].

We here report five unrelated families exhibiting a syndromic form of coloboma associated with homozygous frame-shift mutations in the *FAT1* gene. We demonstrate that *Fat1* knockout mice and zebrafish homozygous for truncating *fat1a* mutations exhibit coloboma, supporting the causality of these mutations and pointing to an evolutionary conserved role of *Fat1* in eye development and optic fissure closure. Furthermore, studies conducted in human primary retinal pigment epithelium (RPE) cells point to a defect in optic fissure margin fusion likely caused by loss of FAT1 at the earliest cell–cell contacts that mediate optic fissure fusion.

## Results

### *FAT1* mutations cause a syndromic form of colobomatous microphthalmia.
We identified homozygous frameshift variants in the atypical protocadherin *FAT1* by whole exome sequencing (WES) and Sanger sequencing confirmation in 10 affected individuals from five unrelated consanguineous families of Middle-Eastern, Turkish, Pakistani, and North-African descent (Fig. 1a, b, Table 1). Patients presented with a previously undescribed syndrome including ocular abnormalities, nephropathy, syndactyly of the toes, and facial dysmorphism (Fig. 1c–i, Table 1). Seven patients presented with bilateral ptosis and two patients had unilateral ptosis (9/10, Fig. 1c). Ocular abnormalities included amongst others microphthalmia (4/10, Fig. 1d) iris coloboma (3/10, Fig. 1e), retinal coloboma (6/10, Fig. 1f, g), and severe amblyopia (5/10). The size of the eye was determined by measuring the axial length of the eye with an echo-biometer. Optical coherence tomography (OCT) images of individual F2-IV-1 are provided in Supplementary Fig. 1. Syndactyly of the toes was seen in 8 out of 10 patients and affected predominantly the 3rd and

4th digits (Fig. 1h). X-ray of the feet demonstrated cutaneous syndactyly (Fig. 1i) in patient F2-IV-1. Patients F3-IV-1 and F3-IV-3 presented with bone fusion of phalanges 3–4 on the right foot and hypotrophy of phalanx 2 of the left foot (Fig. 1h). Dysmorphic facial features included high arched eyebrows, a long philtrum, long nose, and elongated appearance of the face (Fig. 1c). Affected individuals from families 1 and 2 had normal intellectual development corresponding to their age whereas patients F3-IV-3, F4-II-3, and F5-II-1 presented with intellectual disability. Patient F3-IV-1 presented with stage 5 chronic kidney disease at the age of 20 and a biopsy showed focal segmental glomerulosclerosis. His brother, patient F3-IV-3, developed intermittent proteinuria with normal kidney function at the age of 20 years. Patient F5-II-1 was hospitalized at the age of 15 years with proteinuria and hematuria and renal biopsy displayed glomerulotubular nephropathy (Supplementary Fig. 2)[8]. Clinical follow-up of the other patients revealed asymptomatic proteinuria in two siblings from family 1 (patients IV-1 and IV-5).

In all five families mutated alleles were inherited recessively from each of the unaffected consanguineous parents (Fig. 1a). Identified variants were absent in 123,136 exomes and 15,496 genomes from the Genome Aggregation Database (gnomAD, accessed November 2017)[11] and 2497 individuals from the Greater Middle East (GME, accessed November 2017) Variome Project[12]. Furthermore, variant c.2207dupT (p.I737NfsX7) found in Moroccan families 1 and 2 was absent in 400 alleles of individuals of Moroccan descent. All four variants are predicted to results in a premature truncation of the FAT1 protein at amino acids 744, 871, 1043, and 3270, respectively (Fig. 1b). The gnomAD database contains only 46 LoF variants (expected number of LoF variants: 144) and none of them in homozygous condition, in comparison to 2601 missense variants (expected number of missense variants: 2585). We performed two-point linkage analysis for the *FAT1* variants using genotype and pedigree information (Fig. 1). Assuming an autosomal recessive inheritance and a penetrance of 0.99, we obtained a combined maximal LOD score of 5.3 for the *FAT1* variants. Patients F5-II:1 and F4-II-3 had been previously published, of whom the latter was diagnosed with Dubowitz syndrome (OMIM: 223370), although sequencing of *NSUN2* was negative[13].

### Deletion of *Fat1* leads to coloboma in mouse and zebrafish.
Given the important role of FAT1 in forming the earliest cell–cell contacts in epithelia[3] and in maintaining epithelial junctional integrity[3,7,8], we hypothesized that FAT1 is involved in optic fissure fusion via epithelial cell-mediated fusion and thereby underlies the coloboma observed in patients. We analyzed the spatio-temporal mRNA expression of *Fat1* in the developing wild type (WT) mouse eye at E10.5, E11.5, and E12.5, approximately corresponding to "before", "during", and "after" optic fissure closure. *Fat1* had a dynamic expression at the mouse optic fissure margins, periocular mesenchyme, and optic cup with specific spatio-temporal patterns during optic fissure closure (Fig. 2). *Fat1* was expressed in the optic cup, including apposing edges of the optic fissure, in the lens vesicle, and periocular mesenchyme (POM) (Fig. 2a–c). *Fat1* was more highly expressed in the nasal than in the temporal portion of presumptive neural retina (Fig. 2b). Expression of *Fat1* in POM was elevated in the ventral compartment as compared to the dorsal compartment (Fig. 2d–f). The spatio-temporal pattern of *Fat1* expression supports the hypothesis of Fat1 involvement in optic fissure closure. Next, we performed morphological and histological analysis of *Fat1*$^{-/-}$ mouse embryos at embryonic day (E) 14.5 which demonstrated completely penetrant coloboma ($n > 20$), recapitulating the prominent developmental eye defect observed in

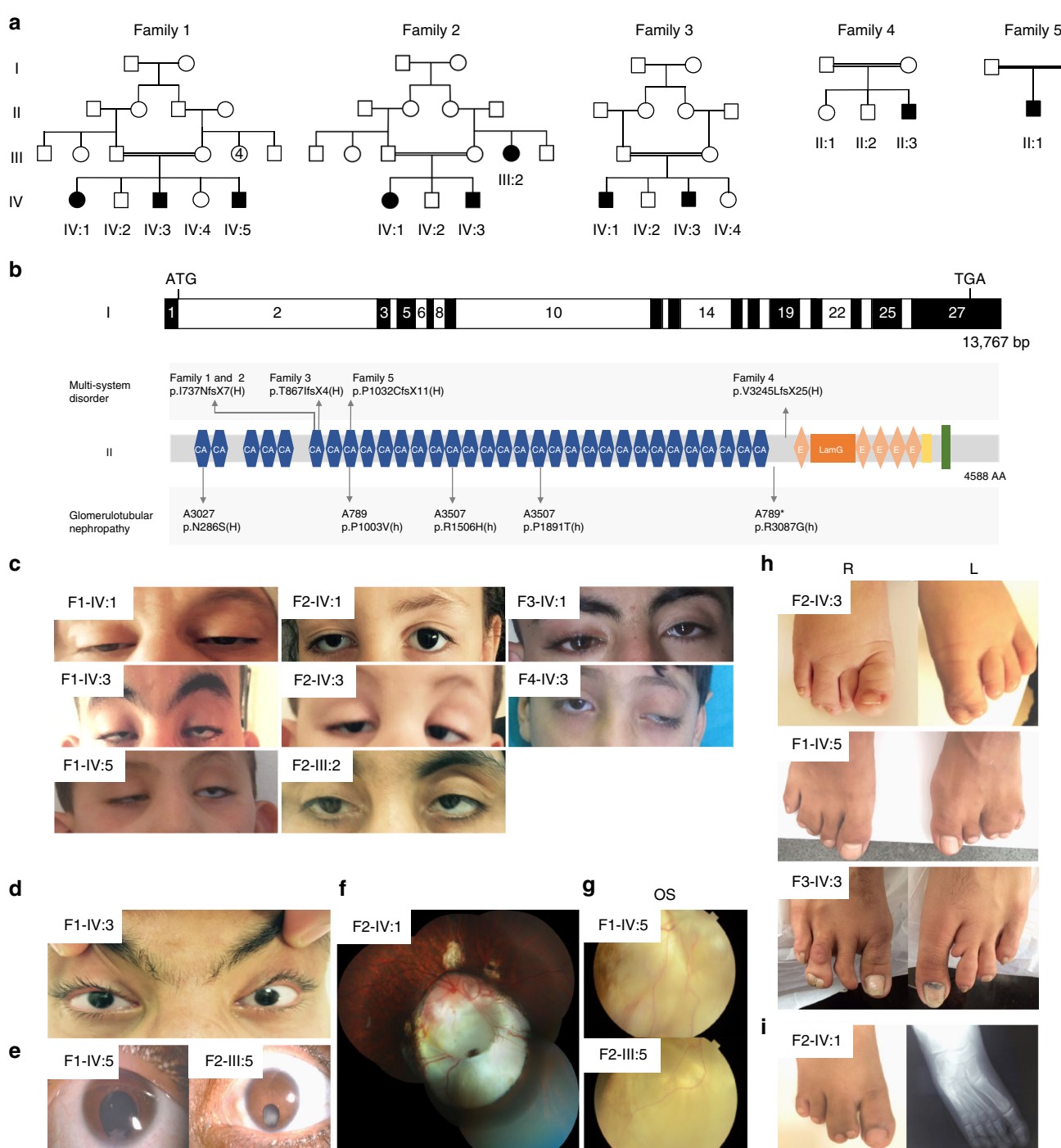

**Fig. 1** Recessive frameshift mutations in *FAT1* cause a new clinical syndrome. Pedigree of family 1–5 (**a**). A schematic of human *FAT1* start/stop codon, exons, location of mutations previous published and identified in this study (top panel). The FAT1 protein is 4588 amino acids long and contains 34 cadherin repeats (CA), followed by five epidermal growth factor (EGF)-like repeat domains (E), a laminin G domain (LamG), a transmembrane domain (green), and an intracellular domain (bottom panel). Family 1 and 2 were found to carry the same homozygous frameshift variant c.2207dupT (p.I737NfsX7) in *FAT1* (NM_005245). In Families 3–5 we identified, respectively, the following homozygous frameshift *FAT1* variants: c.2600_2601delCA (p.T867IfsX4), c.9729del p.(V3245LfsX25), and c.3093_3096del (p.P1032CgsX11). H homozygous, h heterozygous (**b**). Ophthalmic features observed in patients included ptosis; bilateral in patients F1-IV-1, F1-IV-3, F1-IV-5, F2-III-5, F2-IV-3, F3-IV-1 and unilateral in F2-IV-1, F4-II-3 (**c**), microphthalmia (**d**), iris coloboma (**e**), large chorioretinal coloboma containing the papilla/optic nerve with two other smaller circumscribed chorioretinal colobomas localized above the papilla (**f**), and retinal coloboma (**g**). Skeletal abnormalities included syndactyly in the majority of patients and bone fusion of phalanges 3–4 on the right foot and hypotrophy of phalanx 2 of the left foot in patient F3-II-2 (**h**). X-ray of the feet demonstrating cutaneous syndactyly in patient F2-IV-1 (**i**)

**Table 1 Clinical characteristics of homozygous *FAT1* mutations carriers**

| | Family 1 | | | Family 2 | | | Family 3 | | Family 4 | Family 5 |
|---|---|---|---|---|---|---|---|---|---|---|
| | IV:1 | IV:3 | IV:5 | III:2 | IV:1 | IV:3 | IV:1 | IV:3 | II:3 | II:1 |
| *FAT1* mutation (All homozygous) | c.2207dupT (p.I737NfsX7) | | | c.2207dupT (p.I737NfsX7) | | | c.2600_2601delCA (p.T867IfsX4) | | c.9729del (p. V3245LfsX25) | c.3093_3096del (p.P1032CfsX11) |
| Ethnicity | Morocco | | | Morocco | | | Middle-East | | Pakistan | Turkey |
| Consanguinity | + | | | + | | | + | | + | + |
| Sex | F | M | M | F | F | M | M | M | M | M |
| Age (years) | 39 | 34 | 18 | 36 | 8 | 2 | 27 | 24 | 8 | 8 |
| Intellectual disability | − | − | − | − | − | − | − | + | + | + |
| *Ocular features* | | | | | | | | | | |
| Iris coloboma | − | − | B | U | − | B | − | − | − | − |
| Retinal coloboma | B | − | B | B | B | B | − | U | − | − |
| Ptosis | B | B | B | B | U | B | − | B | U | B |
| Microphtalmia | − | − | B | − | U | U | − | U | - | − |
| *Feet abnormalities* | | | | | | | | | | |
| Syndactyly | − | 3rd—4th RF, 3rd—4th LF | 3rd—4th RF, 3rd-5th LF | 3rd—4th RF | 2nd-3rd RF | 1st-2nd LF, 3rd-4th RF | 3rd-4th RF; Bone fusion | 3rd-4th RF; Bone fusion; phalanx hypotrophy | 4th—5th bilaterally | − |
| *Renal manifestations* | | | | | | | | | | |
| Nephropathy | + | − | + | − | − | − | + | + | − | + |
| Biopsy (at age) | n/a | n/a | n/a | n/a | n/a | n/a | FSGS (20 years of age) | n/a | n/a | TIN, MS, thin GBM (12 years) |

For a more detailed phenotype description see Supplementary Table 2
*B* bilateral, *F* female, *FSGS* focal segmental glomerulosclerosis, *GBM* glomerular basement membrane, *LF* left foot, *M* male, *MS* mesangial sclerosis, *n/a* not assessed, *U* unilateral, *RF* right foot, *TIN* tubular interstitial nephritis; + phenotype present, − phenotype absent

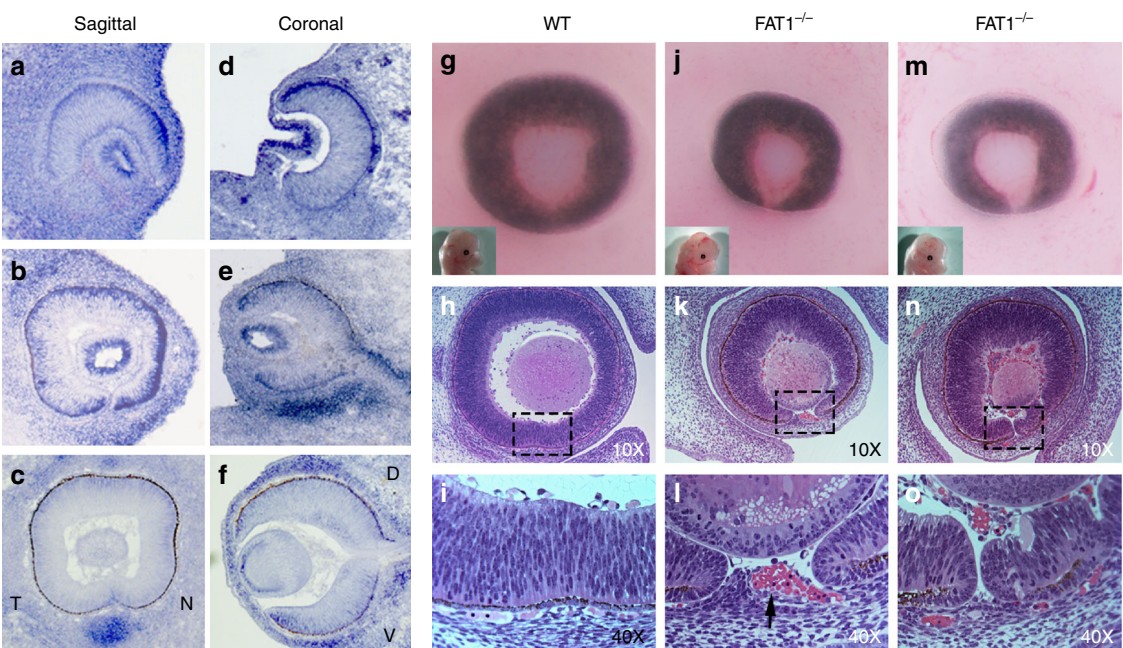

**Fig. 2** *FAT1⁻ᐟ⁻* mouse exhibit microphthalmia and completely penetrant coloboma. *FAT1* mRNA expression pattern in sagittal (**a–c**) and coronal (**d–f**) mouse optic cup sections at E10.5 (**a, d**), E11.5 (**b, e**), and E12.5 (**c, f**). Compared to wild-type (WT) embryos (**g–i**, E14.5) *FAT1⁻ᐟ⁻* mouse exhibited microphthalmia (**j, m**). Fused optic fissure margins of WT mouse embryos (E14.5, **h, i**; sagittal section) and unfused margins were seen in *Fat1⁻ᐟ⁻* mice (**n, o**) and/or persistence of POM—including early hyaloid vessel precursors (**k, l**, arrow) interposing between the closing edges of the fissure (E14.5). D dorsal, V ventral, T temporal, N nasal

affected individuals. At E14.5, when the optic fissure is completely fused in WT (Fig. 2g–i), *Fat1⁻ᐟ⁻* mice showed unfused margins (Fig. 2n, o) and/or persistence of POM—including early hyaloid vessel precursors (Fig. 2k, l, arrow) interposing between the closing edges of the fissure. Compared to wild-type (WT) embryos (Fig. 2g) *FAT1⁻ᐟ⁻* mouse exhibited microphthalmia (Fig. 2j, m). The coloboma phenotype was observed to be completely penetrant in the homozygous mutant mice and was never observed in heterozygous mutant mice. Occurrence of unfused optic fissure margins apposed to each other and presence of POM tissue in the choroid fissure obstructing the fusion of optic fissure margins, suggests multiple mechanisms like fusion and morphogenetic defects contributing towards the coloboma phenotype.

In zebrafish, *fat1a* expression was observed to be restricted to the rostral end of the embryo (Supplementary Fig. 3a–d) during the time of optic fissure closure (24-h post-fertilization), as has been reported previously[7]. Morpholino-mediated knockdown of the zebrafish *fat1a* using two different concentrations, consistently resulted in coloboma (Supplementary Fig. 4a–d), which was further confirmed by detailed histological analysis of day 3 post-fertilization larvae (Supplementary Fig. 4e–h). At lower doses of morpholino the coloboma phenotype observed was similar to that observed for mouse, where the two optic fissure margins were very close to each other but failed to fuse. The variability in morphant phenotype observed at high doses of morpholino are depicted in Supplementary Fig. 5.

**FAT1 is required for junctional integrity in human RPE cells.** To further investigate the role of *FAT1* in optic fissure closure, we studied human primary RPE, the cell type present at the leading edge of optic fissure margins that mediates fissure closure[14]. The fusion of optic fissure margins is initiated by cellular processes emanating from the apposing edges of optic fissure margins, observed using transmission electron microscopy (TEM) (Supplementary Fig. 6a), and forming "simple appositional type," contacts[15]. The failure to fuse optic fissure margins represents a

classical epithelial fusion defect during eye development[10]. Using confocal microscopy, we observed FAT1 immunostaining with F-ACTIN fibers at the leading edges and filipodia of isolated RPE cells (Supplementary Fig. 6b–d), as has been reported for other epithelial cell types. The specificity of the FAT1 antibody used was validated using mouse embryonic fibroblast (MEF) isolated from WT and *Fat1*$^{-/-}$ mouse embryos (Supplementary Fig. 6c), where FAT1 immuno-staining was observed at the earliest cell–cell contacts in WT cells but was absent in FAT1$^{-/-}$ cells. In sub-confluent RPE cultures, FAT1 localized with F-ACTIN fibers at leading cell edges (Supplementary Fig. 6b, d) and with ZO-1 at the earliest cell–cell interactions (Supplementary Fig. 6e), respectively.

We usually culture RPE cells on a 2D trans-well culture system where they form a polarized monolayer, to recapitulate in vivo electro-physiological properties of RPE monolayer. In confluent monolayer polarized RPE cultures, FAT1 localized at the cell borders similar to the organized F-ACTIN filaments and ZO-1 staining (Fig. 3b, c top panels) suggesting that it might be involved in formation or maintenance of cell–cell junctional complexes. Short hairpin RNA (shRNA)-mediated knockdown of *FAT1* in sub-confluent RPE cultures resulted in loss of FAT1 from filopodia and disruption of ZO-1 immuno-staining at the

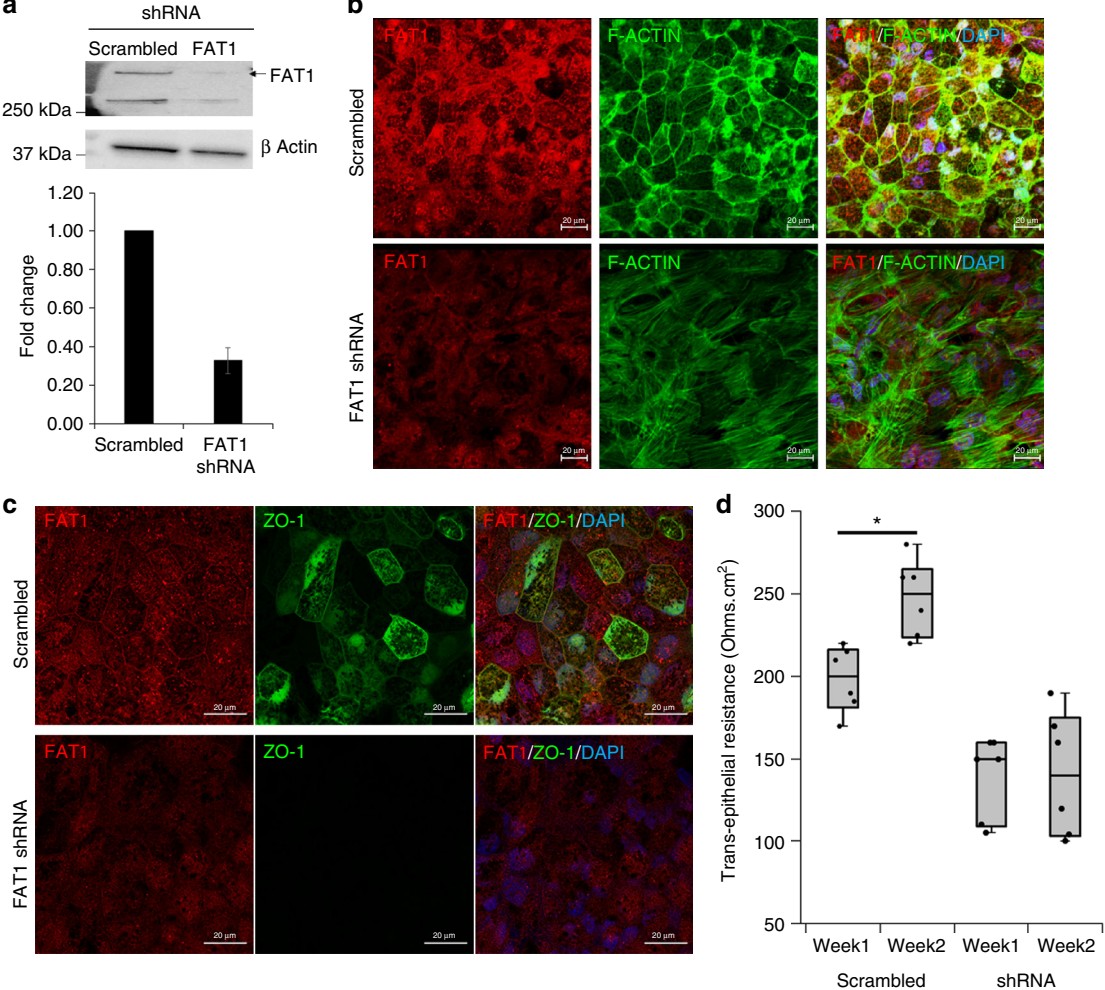

**Fig. 3** FAT1 knockdown disrupts F-ACTIN and ZO-1 localization in RPE. ShRNA-mediated knockdown of FAT1 in RPE cells was confirmed using Western blotting (**a**, N = 3). Disorganization of F-ACTIN fibers (**b**) and loss of ZO-1 staining pattern (**c**) was observed in RPE cells upon knockdown of FAT1. Trans-epithelial electrical resistance was measured at week one and two post-FAT1 knockdown to determine junctional integrity (**d**, N = 6). Scale bar is 20 μm. Error bars represent standard error of mean, and statistical significance is at $P < 0.05$ (two tailed *t*-test) denoted by "*". Raw data is provided in supplementary section

earliest cell–cell junctions (Supplementary Fig. 6d, e). Knockdown of *FAT1* in RPE cells before monolayer formation (Fig. 3a) resulted in failure to organize F-ACTIN fibers (Fig. 3b), loss of ZO-1 staining from cell–cell junctions (Fig. 3c), and significantly reduced trans-epithelial barrier potential, suggesting compromised junctional integrity (Fig. 3d). As such, FAT1 might play an important role in the earliest RPE cell–cell junctions and F-ACTIN filament organization during RPE monolayer formation. To further confirm our assumption that junctional integrity is compromised due to FAT1 knockdown we studied β-CATENIN staining which is known to localize at adherens junctions. Although not as dramatic as loss of ZO-1 staining, we observed a significant disruption of β-CATENIN staining (Supplementary Fig. 7), confirming our assumption that FAT1 might be involved in establishing junctional integrity of RPE monolayer. These observations were further supported by TEM studies where shRNA-mediated knockdown of FAT1 resulted in a failure to organize RPE cells into an epithelial monolayer (Fig. 4a, b) and disrupted epithelial junctions (Fig. 4c, d). We currently do not know if the failure to organize RPE cells into an epithelial monolayer is primarily due to loss of FAT1 or a secondary defect resulting from disrupted junctions.

Interestingly, in vivo differentiation of RPE from neuro-ectodermal to an epithelial cell type is not affected by loss of FAT1 as observed during mouse eye development (Fig. 2g–m). No defects were observed in the organization of F-Actin fibers and ZO-1 staining pattern in the RPE flat-mounts from WT and $Fat1^{-/-}$ mouse embryos (Supplementary Fig. 8), suggesting a specific role of FAT1 in epithelial sheet fusion by mediating

earliest cell–cell junctions, rather than differentiation towards epithelial lineage.

**The Fat1 cytoplasmic tail is involved in optic fissure closure**. We also studied the expression of the remaining *Fat* gene family members, *Fat2*, *Fat3*, and *Fat4*, in the developing WT mouse eye at E10.5, E11.5, and E12.5. *Fat2*, and *Fat3* were not expressed in the developing eye. The *Fat4* mRNA expression pattern was overlapping with that of *Fat1* (Supplementary Fig. 9a–e). The *Fat4* mRNA expression was observed around the optic cup in peri-ocular mesenchyme and RPE. Although, we observed that both *Fat1* and *Fat4* had overlapping expression pattern in the developing mouse eye, we only found *Fat1* to be essential for optic fissure closure in mouse and zebrafish. Unlike $Fat1^{-/-}$ mice, no developmental defects, including microphthalmia or coloboma were observed in $Fat4-/-$ mice eyes by E12.5 ($n = 20$, Supplementary Fig. 9g–l). Similarly, morpholino-mediated knockdown of the zebrafish *fat4* showed no apparent defects during eye development. A similar observation was made during zebrafish pronephros and lens epithelia development, where morpholino-mediated *fat1* knockdown, but not *fat4*, resulted in cystic pronephros[7] and loss of apical–basal polarity of $Fat1^{-/-}$ mouse lens epithelia[6]. This is in line with the observation that recessive mutations in *FAT4* result in Van Maldergem syndrome (MIM615546) and Hennekam lymphangiectasia-lymphedema syndrome 2 (MIM616006), neither of which include coloboma or any other eye defects.

Since FAT1 and FAT4 have similar extra-cellular domains, but diverge significantly in the cytoplasmic region, we hypothesized that the FAT1 cytoplasmic tail could be involved during optic fissure closure. We therefore targeted the cytoplasmic tail region of zebrafish *fat1a* using CRISPR/*cas9* and selected for frameshift truncating alleles, resulting in loss of the two preferred VASP/MENA and the PDZ-binding domains located at the C-terminal end, due to a premature stop codon (Supplementary Fig. 10a, b). We observed optic fissure fusion defects in embryos with homozygous alleles of truncated *fat1a* (Fig. 5a, b, e, f). This was further confirmed by detailed histological analysis of optic cup during the time of optic fissure fusion (Fig. 5c, d, g and h). Sagittal section through the zebrafish optic cups at the time of optic fissure fusion revealed abnormal organization of the optic fissure margins and migration and/or delayed differentiation of the RPE cells at the optic fissure margins (Fig. 5d, h, black arrows) in homozygous *fat1a* mutants. Compromised structural integrity of RPE monolayer around the optic cup of homozygous *fat1a* mutants was also observed, pointing towards a combination of morphogenetic and structural organizational defects as the cause of fissure margins fusion failure. These observations also suggest that VASP/MENA and PDZ-binding domains located in the cytoplasmic tail of FAT1 are involved in optic fissure closure.

## Discussion

We here identified five unrelated families presenting with a new syndrome consisting of colobomatous microphthalmia, ptosis, and cutaneous syndactyly with or without glomerulotubular nephropathy, associated with homozygous frame-shift mutations in the *FAT1* gene. The causal relation between loss of function mutations in *FAT1* and coloboma was established through studies in mice and fish loss-of-function models, underscoring a highly conserved role of *FAT1* during vertebrate eye development. We demonstrated for the first time that FAT1 is expressed in human primary RPE cells, the primary site for optic fissure margins fusion, placing it exquisitely at the precise cell type and location for facilitating optic fissure fusion during vertebrate eye development. Using an in vitro culture system, we showed that FAT1

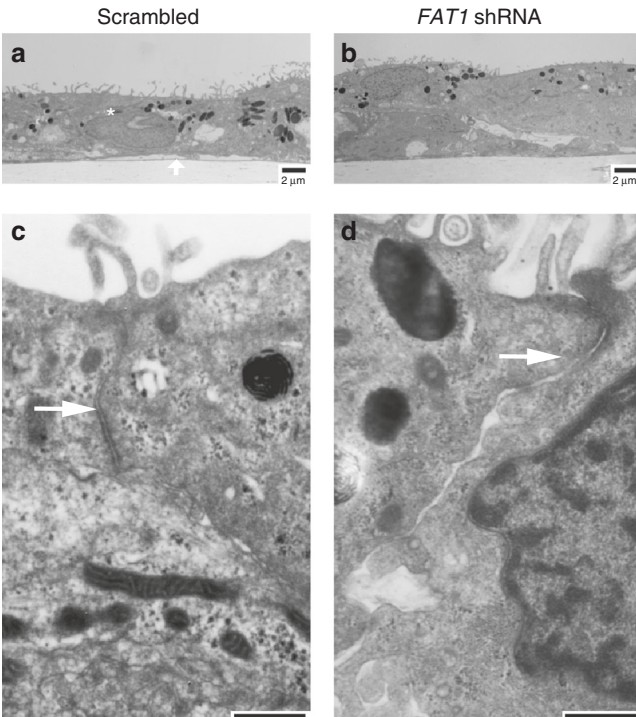

Scrambled                    *FAT1* shRNA

**Fig. 4** FAT1 knockdown results in a failure to organize RPE cells into a monolayer. Transmission electron miscroscopy (TEM) of RPE cells cultured on trans-wells for 2 weeks following treatment with scrambled (**a**, **c**) and FAT1 shRNA (**b**, **d**). A transverse section depicting RPE cells (*, scale bar is 2 μm) sitting on top of trans-well membrane (**a**, arrow). A higher magnification showing epithelial tight junction (**c**, **d**, arrow, scale bar is 500 nm) between two neighboring cells

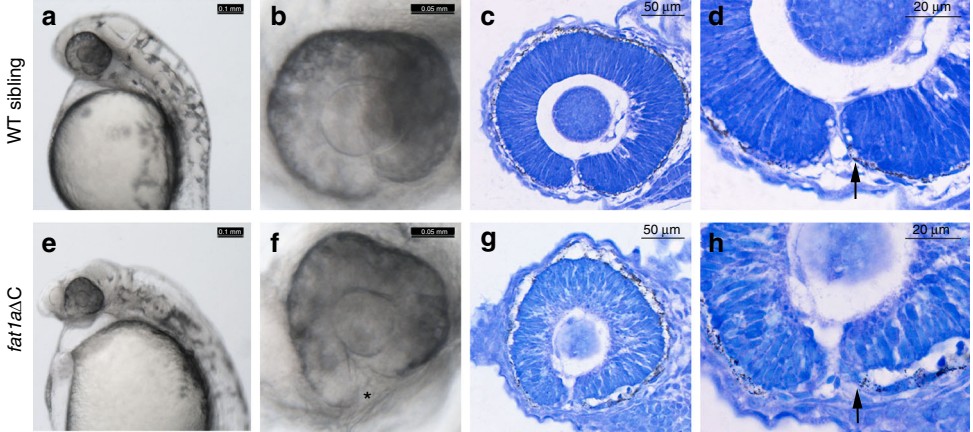

**Fig. 5** Zebrafish embryos with homozygous alleles of truncated *fat1a* display coloboma. CRISPR/Cas9-mediated introduction of frame-shift mutations in FAT1 C-terminal resulted in optic fissure closure defects (**a** and **e**, scale bar is 0.1 mm). A higher magnification of eye depicting fused margins in WT and unfused margins in homozygous mutant (*, **b** and **f**, scale bar is 0.05 mm). Sagittal sections of zebrafish embryos (24–30 hpf) followed by toludene blue staining showing organization of the optic cup (**c** and **g**, scale bar is 50 μm). Higher magnification of the optic cup shows morphology of optic fissure margins in WT and homozygous mutant (**d**, **h**, scale bar is 20 μm)

is essential for establishment of the earliest cell to cell junctions, F-Actin organization, and maintenance of junctional integrity of RPE monolayer.

The novel clinical syndrome we describe here joins the growing list of human conditions involving both coloboma and renal disease. The differential diagnosis includes amongst others CHARGE, Papillorenal, Temtamy, and Klippel–Feil syndromes. Mutations in *SALL1/SALL2*, *YAP1* and *STRA6* also result in a syndromic form of coloboma with a renal phenotype[8,16]. Although we found a strikingly high penetrance of ocular and limb defects in affected individuals, renal dysfunction was less prominent. We found a wide spectrum of renal involvement ranging from asymptomatic proteinuria (3/10) to severe nephrotic syndrome (2/10). In addition, in some of the affected individuals, proteinuria was only present intermittently. We therefore recommend that asymptomatic *FAT1* patients should undergo clinical evaluation of kidney function, which should be repeated on a regular basis. *FAT1* is another example of a gene wherein genetic variation has pleiotropic effects, with biallellic missense variants leading to non-syndromic glomerulotubular nephropathy and homozygous frameshift variants causing the broad multi-system disorder that we describe here.

Our mechanistic studies in mice, zebrafish, and RPE cell cultures support the causality of *FAT1* biallelic frameshift variants in the pathogenesis of coloboma, and for the first time implicate *FAT1* in optic fissure fusion during eye development. During optic fissure closure apposing margins of the presumptive neural retina that are lined by presumptive RPE adhere initially by cellular protrusions (Supplementary Fig. 6a), then proceed to form more stable appositional junctions[17] and subsequently undergo differentiation of the RPE and neuroepithelium to yield a complete eye where RPE is overlying neural retina. In our previous study we showed that genes that are dynamically expressed at the apposing edges of the optic fissure margins during the process of optic fissure closure are important in this developmental process[18]. Tissue collected from optic fissure margins in wild-type mouse embryos at E10.5, E11.5, and E12.5—approximately corresponding to "before", "during", and "after" optic fissure closure—identified the important role of *Nlz* genes in optic fissure closure in mouse and zebrafish[18]. In the same screen we independently identified *FAT* atypical cadherins as potential candidates for optic fissure closure[18]. Though we were not able to show FAT1 immunoreactivity at the earliest cellular processes that

initiate the optic fissure fusion process, due to technical difficulties, we did observe the presence of Fat1 mRNA at the leading edges of the optic fissure margins (Fig. 2b, c, e, f). In cell culture experiments we observed FAT1 immunoreactivity at leading edges, filopodia, and earliest cell–cell contacts of RPE cells (Supplementary Fig. 6b–e). Neural tube closure, like optic fissure fusion, is a classical epithelial sheet adhesion and fusion morphogenetic event. *Fat1*[−/−] mice display partially penetrant anterior neural tube closure defects due to reduced actin accumulation, leading to aberrant apical constriction in the neural epithelium cells[9]. The presence of optic fissure and neural tube closure defects supports the role of *FAT1* in epithelial cell–cell adhesion and/or fusion

We demonstrate that FAT1 plays an essential role in formation and maintenance of junctional integrity. The observations that support this finding include disruption of ZO-1 and β-CATENIN immuno-staining at the earliest cell–cell junctions, failure to organize F-ACTIN fibers and failure of RPE cells to organize an epithelial monolayer resulting in reduced trans-epithelial barrier potential. Disrupted epithelial junctions suggests a critical role of FAT1 in formation and maintenance of earliest cell–cell junctions. In line with the ocular and kidney involvement in affected individuals, loss of FAT1 results in decreased cell adhesion in podocytes (an epithelial cell type), and disrupted lumen formation in renal tubular cells[8].

Our work in zebrafish where we targeted the cytoplasmic tail region of zebrafish *fat1a* using CRISPR/*cas9* provides evidence that VASP/MENA and PDZ-binding domains located in the cytoplasmic tail of FAT1 are involved in optic fissure closure. The FAT1 cytoplasmic tail acts as a binding partner for β-CATENIN, RERE, ENA/VASP, and PDZ domain containing proteins including SCRIBBLE. Previous studies have suggested the association of coloboma with β-CATENIN[19], RERE[20], and SCRIBBLE[21]; currently we do not know which of these proteins are involved in *FAT1*-mediated coloboma. The ENA/VASP proteins have been implicated in F-ACTIN polymerization/reorganization during epithelial adhesion[22], and establishing FAT1-mediated initial cell–cell contact[3,23].

While a failure to fuse the fissure margins during the process of optic fissure fusion is a commonly appreciated cause for coloboma formation, and our data supports such a mechanism, it could also be due in part to morphogenetic events. Our finding that intervening peri-ocular mesenchyme can be observed

obstructing fusion of fissure margins (Fig. 2k, l panels) also provides evidence for an underlying morphogenic event that may at least in part explain the coloboma phenotype. We acknowledge that further work is needed to provide more detailed insight into the latter mechanism.

Heterozygous loss of function mutations in human *YAP1* and loss of *Cdc42* in mouse have been associated with ocular coloboma[24]. These two genes have both been implicated in Hippo signaling[25]. Yes-associated protein (YAP) is a downstream transcriptional co-activator inhibited by Hippo signaling has a crucial role in nephron induction and morphogenesis[25], and specification of RPE cell fate during eye development[14]. Cdc42 has been shown to act upstream of YAP in nephron progenitor cells to promote YAP-dependent gene expression necessary to form functioning nephrons[25]. Tissue-specific deletion of Yap1 as well as Cdc42 in mouse results in strikingly similar and severe defects in kidney development[25]. The PDZ-binding domain of FAT1 has been shown to recruit Scribble for mediating Hippo signaling that leads to Yap inhibition, during pronephros development in zebrafish[7]. Recently, loss of FAT1 has been shown to decrease Cdc42 activity[8]. Furthermore, in vitro studies in HEK293 cells showed that FAT1 knockdown results in nuclear accumulation of YAP1[26]. Together, these data suggest that the ophthalmic phenotype due to loss of FAT1 may also result from deregulated YAP1 activity. However, shRNA-mediated knock-down of FAT1 in primary human RPE monolayer on trans-wells showed no significant differences in YAP/TAZ localization pattern compared to scrambled shRNA control (Supplementary Fig. 11a). Also, protein levels ($n = 3$) of total YAP/TAZ and phosphorylated form of YAP (pYAP-Ser127) were unchanged in scrambled and *FAT1* shRNA-treated RPE monolayers (Supplementary Fig. 11b). Similarly, no differences were observed in YAP/TAZ-staining patterns between MEF cells isolated from WT and $Fat1^{-/-}$ mouse embryos (Supplementary Fig. 11c). Moreover, YAP immunostaining in mouse E11.5–12.5 optic cup sections (sagittal) identified no differences in the pattern of YAP staining between WT and $Fat1^{-/-}$ mouse presumptive neural retina and RPE cells (Supplementary Fig. 11d), While we did not observe any significant changes in YAP levels and localization upon loss of FAT1 in our experiments, additional studies on mechanisms by which FAT1 regulates YAP1 during eye development are warranted. Since we observed *FAT1* RNA labeling in POM it is possible that FAT1 could be involved in regulating YAP levels in the POM but not in the RPE.

In conclusion, our study associates *FAT1* frameshift mutations to a previously undescribed clinically recognizable syndrome consisting of ophthalmic, facial, renal, and skeletal abnormalities. *Fat1* ablation in mice and zebrafish showed a similar ophthalmic phenotype, for the first time underscoring the fundamental and evolutionary conserved role of the atypical cadherin *FAT1* during optic fissure closure. FAT1 also seems to be involved for earliest cell–cell contacts, apical junction formation, and F-ACTIN reorganization in primary human RPE cells. Our study identified *FAT1* as another example of a gene wherein genetic variation has pleiotropic effects, where biallellic missense variants lead to nonsyndromic glomerulotubular nephropathy and homozygous frameshift variants lead to a broad multi-system disorder.

## Methods
**Clinical evaluation and WES**. Patients were referred to the Department of Clinical Genetics for further comprehensive genetic analyses and counseling. The study protocol was approved by the local Institutional Review Boards where the patients were followed and signed informed consent was obtained from the patients or their parents. The authors affirm that human research participants provided informed consent, for publication of the images in Fig. 1. Sequencing protocols and mean coverage for each family is summarized in Supplementary Table 1.

For family 1, genome-wide homozygosity mapping was performed in patients IV-1, IV-3, IV-5, and their parents, by using the Affymetrix 250K NspI SNP genotyping microarray. Multipoint LOD scores across the whole genome were calculated using the MERLIN software (http://www.sph.umich.edu/csg/abecasis/Merlin), assuming recessive inheritance with complete penetrance. Three regions of homozygosity were shared by the three affected siblings: on chromosome 4 with a size of 9.5 Mb, on chromosome 13 with a size of 1 Mb, and on chromosome 17 with a size of 2.2 Mb. Because of the high number of genes present in these regions, WES was then conducted in patients IV-3 and IV-5, and their parents using Agilent SureSelect libraries and sheared with a Covaris S2 Ultrasonicator. Exome capture was performed with the 51 Mb SureSelect Human All Exon kit v5 (Agilent Technologies). Sequencing was carried out on a pool of barcoded exome libraries using a HiSeq 2500 instrument (Illumina), generating 100 + 100 bp paired-end reads, with a mean depth of coverage of ×140. After demultiplexing, paired-end sequences were mapped to the reference human genome (GRCh37/hg19 assembly, NCBI) using Burrows–Wheeler aligner (BWA). Downstream processing was performed using the Genome Analysis Toolkit (GATK)[17], SAMtools[27], and Picard. Variant calls were made with the GATK UnifiedGenotyper. Variant annotation was based on Ensembl release 71. Variants were filtered against publicly available SNPs plus variant data from more than 7000 in-house exomes (Institut *Imagine*).

In family 2, the coding exons were captured using the Agilent SureSelect Human All Exon v5 and sequenced on the Hiseq2000 sequencer (Illumina) at a target mean depth of 50 reads per target base. SOAPaligner (version 2.21) was used to align sequence reads to the human reference genome (NCBI build 37/hg19). For single nucleotide variants, functional annotation was performed with SOAPsnp. For insertion/deletion detection, we aligned sequence reads by BWA and performed annotation with GATK.

In family 3, patient genomic DNA was isolated, followed by exome capture using the SureSelect Human All Exon kit V4 (Agilent technologies Inc). Sequencing of captured fragments was performed on the HiSeq2500 platform with a mean depth of coverage of 84X After alignment of sequence reads, the DNAnexus software (Palo Alto, CA, release October 2011) was used to perform variant calling with the human reference genome (NCBI build 37/hg19) as reference. Variant filtering was subsequently performed against public databases (dbSNP129, HapMap, 1000 Genomes).

In family 4, exome capture for WES was performed using the NimbleGen SeqCap EZ Human Exome Library (Roche NimbleGen Inc., Madison, WI). Sequencing of paired-end 100 bp fragment reads was performed on the Illumina HiSeq2000 with a mean depth of coverage of ×90. The VarBank v.2.1 data analysis pipeline was used for mapping, alignment, and variant calling. Alignment of sequenced reads, indexing of the reference genome, variant calling, and annotation was conducted using a pipeline consisting of BWA, Samtools, Picard, and Annovar[28].

In family 5, genomic DNA was isolated from blood lymphocyte or saliva samples. Exome capture for WES was performed using the SureSelect Human All Exon kit V5 (Agilent technologies Inc). Subsequently, captured fragments were sequenced on the llumina HiSeq2000 with a mean depth of coverage of ×50. The CLC Genomics Workbench™ (version 6.5.2) software (CLC bio, Aarhus, Denmark) was used to align sequence reads to the human reference genome (NCBI build 38/hg19). Variant filtering was then performed against public databases (dbSNP147 and 1000 Genomes). We excluded synonymous and intronic variants that were not located at splice sites. WES data was evaluated for potential disease-causing mutations in >50 known monogenic genes associated to nephrotic syndrome. A second round of filtering of the remaining variants was conducted using public databases (EVS server, ExAC, gnomAD and 1000 Genomes). Variants identified in families 1–5 were validated using Sanger sequencing and whenever parental DNA was available, segregation analysis was performed. A combined LOD score across the families was calculated using phenotype, genotype, and pedigree information of each family by two-point linkage analysis with the use of the Superlink-Online software package[29] with the assumption of a recessive pattern of inheritance, a disease-allele frequency of 0.001 and a penetrance of 0.99.

**Genotyping of *FAT1* c.2207dupT in Moroccan controls**. Blood samples from the umbilical cord were collected from 200 unrelated newborns. Individuals originated from different regions in Morocco and the Moroccan origin of parents and grandparents was established. DNA was extracted from 3 mL blood with standard salting-out method. Informed consent was obtained from the parents. We developed a realtime polymerase chain reaction (PCR; Applied Biosystems 7500 Fast Real-Time PCR Systems) assay using TaqMan probes for the *FAT1* c.2207dupT mutation (Table 1), and validated the assay using homozygous and heterozygous members of Family 1.

**Animal breeding and maintenance**. All animal studies were performed in accordance with the NEI/NIH animal ethics committee guidelines (NEI-605 and NEI-648). C57BL/6J mice were obtained from Jackson Laboratories (Bar Harbor, ME). *Fat1* and *Fat4* knockout mice have been described previously[6] and were housed and bred according to an NEI-approved animal study protocol (NEI-605). Embryo harvesting was performed according to standard protocol, with E0.5 marking the day after the appearance of a cervical mucus plug. Adult zebrafish were maintained in an automated fish housing systems following a standard

protocol and zebrafish embryos were maintained at 28.5 °C in fish embryo medium as previously described[30].

**Histological and in situ staining of mouse tissue**. Mice were euthanized using carbon dioxide followed by cervical dislocation. Isolated mouse embryos for histopathology were dissected on ice-cold phosphate buffered saline (PBS) and fixed overnight in phosphate-buffered 4% paraformaldehyde at 4 °C. Methacrylate sections through the optic fissure (eyes) or in appropriate sagittal orientation, stained with hematoxylin and eosin (H&E), were used for histopathology. In situ hybridization was performed using a Fat4 probe that included the 3′ UTR cloned into a plasmid as previously described[4,31]. The plasmid for generating the Fat1 probe (ID 6841037) was from Open Biosystems (Lafayette, CO, USA). In situ and immuno-fluorescence staining of mouse cryo-sections was performed as previously described[32]. For RPE flat-mounts, mouse embryonic (E14.5–15.5) eye cups were isolated and dissected to remove presumptive neural-retina. The surrounding choroidal tissue was not removed completely to provide structural support to the RPE monolayer. The dissected out RPE was immuno-stained with ZO-1 and F-Actin as described below. The plane of RPE during confocal imaging of flat-mounts was determined by adding a transmitted PMT (T-PMT) to visualize the location of melanosomes present on the apical side of RPE.

**Zebrafish embryo manipulation by morpholino and CRISPRcas9**. Previously described zebrafish *fat1a* translation-blocking morpholino oligonucleotides[7,8] (Gene Tools LLC., Philomath, OR), diluted in 0.1 M KCl, nuclease-free water and phenol red, were injected (0.45 and 0.52 pMol) in freshly fertilized zebrafish embryos at the one cell stage. The knockdown experiments were repeated using a different morpholino *fat1a* (TTTGCAGCGCACTCCTCTCTGAAAC) to validate the results. The zebrafish *fat4* morpholino (CCGGGTTTTCCCGAGCCTCATAC AT) which has been previously reported[7], and the injection control morpholino (CCTCTTACCTCAGTTACAATTTATA), were used as described above for the *fat1a* morpholino.

CRISPR targets and primers spanning the targets were selected using the online tool CHOPCHOP[33]. The selected target had no off-target sequences of homology. The oligonucleotides containing T7 promoter, target, guide RNA scaffold, and termination signal sequences were purchased from IDT (Coralville, IA) in the form of gBlocks® fragments. Guide RNA were synthesized in vitro by driving transcription mediated by T7 promoter using the HiScribe™ T7 High Yield RNA Synthesis Kit (NEB, Ipswich, MA). For targeting the cytoplasmic tail of zebrafish *fat1a*, ~1 nL of in vitro synthesized guide RNA (200 ng/μL) and Cas9 protein (500 ng/μL) were injected in one-cell stage embryos (F0 embryos). F0 embryos were maintained until adulthood and then outcrossed to WT ABTL to isolate F1 progenies carrying frame-shift mutations in the germline. The selected F1 progenies were inbred to generate F2 embryos that were screened for coloboma phenotype and confirmation by genotyping. Zebrafish *fat1a* mutants were maintained as heterozygotes and inbreed to generate homozygous mutants.

**Cell culture experiments**. Mouse embryonic fibroblast cells were derived from embryonic day 12.5 (E12.5) embryos as described previously[34]. Primary human fetal RPE cells kindly provided by Dr. Sheldon Miller, derived from cadaver eyes obtained from Advanced Bioscience Resources (Alameda, CA, USA). RPE cells were cultured in Primaria® tissue culture flasks (BD Biosciences, Franklin Lakes, NJ) in culture medium based on MEM-α, as described previously[35]. Once confluence was reached, cells were seeded onto vitronectin-coated polystyrene-based trans-wells placed in the wells of a 12-well plate or four-well chamber slides for shRNA-mediated knockdown, immuno-fluorescence staining, and confocal microscopy, as described previously[36]. FAT1 (SHCLNV-NM_005245) and scrambled (SHC016H) shRNA lentiviral transduction particles were purchased from Sigma Aldrich (St. Louis, MO, USA). Lentiviral transduction was performed 48 h post seeding on four-well chamber slides followed by fixation and immuno-fluorescence staining 48 h post transduction. In trans-well culture system, shRNA lentiviral transduction was performed 48 h and again at 120 h post seeding, followed by electron microscopy and immunofluorescence staining 2 weeks post transduction. Trans-epithelial resistance of RPE monolayers on trans-well cultures was measured using EVOM2 (World Precision Instruments, Sarasota, FL, USA) at 1 and 2 weeks post-shRNA transduction, as described by manufacturer (https://www.wpiinc.com/var-2754-epithelial-volt-ohm-teer-meter). For TEM, mouse eyes (E11.5) and RPE monolayers on trans-wells (2 weeks post transduction) were processed as previously described[37]. Briefly, specimens were doubly fixed in gluteraldehyde (2.5% in PBS) and osmium tetroxide (0.5% in PBS), dehydrated, and embedded in Spurr's epoxy resin. Ultrathin sections (90 nm) were prepared and double-stained with uranyl acetate and lead citrate, and viewed with JEOL JEM-1010 (Peabody, MA, USA) and photographed.

**Immunofluorescence staining of cells and confocal imaging**. Cultured cells were fixed for 15 min in 4% paraformaldehyde (PFA) in PBS. After washing with 1× PBS and permeabilization and blocking in ICC buffer (0.5% BSA, 0.5% Tween, and 0.1% triton X100 1× PBS). Cells were then incubated overnight at 4 °C with the primary antibody in ICC buffer (1:100 dilution). After multiple washes in PBS, the cells were incubated for 1 h at room temperature in Alexa488 or 555 conjugated

goat anti-rabbit and/or ant-mouse antibody or Phalloidin (Abcam ab176756, ab176753) and Hoechst33342 (1:1000 dilution in ICC buffer). Cells were then washed in PBST before mounting with Fluoromount-G® (SouthernBiotech, Birmingham, AL, USA) imaging. Primary antibodies used were FAT1 (Abcam: ab190242 and EMD Millipore: MABC612), YAP/TAZ (Cell Signaling Technology: 8418), FITC-tagged ZO-1 (Invitrogen: 33–9111). FAT1 antibody specificity was confirmed in mouse embryonic fibroblast derived from WT and *Fat1*$^{-/-}$ mouse.

Zeiss confocal microscopes 700 and 880 coupled with Airyscan® detector was used for confocal imaging. The images were analyzed using ZEN Software (Carl Zeiss Microscopy LLC, Thornwood, NY). The cell culture experiments were repeated three times for each scrambled and FAT1 shRNA conditions.

**Western blotting**. For Western blotting cells were lysed with RIPA lysis buffer (Sigma-Aldrich, St. Louis, MO) containing protease inhibitor cocktail and Halt™ phosphatase inhibitor cocktail (Pierce Biotechnology), centrifuged at 14,000 × *g*, and supernatant was collected. Total protein concentration was determined by BCA protein assay kit (Pierce Biotechnology, Foster City, CA, USA). Samples (40 μg/each well) were SDS–PAGE electrophoresed using Criterion XT system for high molecular weight proteins (XT-running buffer, XT-sample loading dye and 3–8% Tris–acetate pre-casted gels Bio-Rad, Hercules, CA, USA), blotted onto polyvinylidene fluoride (PVDF) membranes (Bio-Rad) and immunoreacted with antibodies to FAT1 (Abcam: ab190242), YAP/TAZ (Cell Signaling Technology: 8418), phosphor YAP- Ser127 (Cell Signaling Technology:13008), and β-ACTIN (Cell Signaling Technology: 3700) for normalization (1:1000 dilution). The membranes were then incubated with secondary antibodies (LI-COR, goat anti-mouse IRDye 800CW or donkey anti-rabbit IRDye 680RD, 1: 20,000 dilution) for 45 min at room temperature followed by washings, scanned using the Odyssey infrared scanner and analyzed using Image Studio Lite Ver.4.0 (LI-COR Inc., Lincoln, NE, USA). Uncropped scan of the blots is provided in the Supplementary Information section.

**URLs**. For dbSNP, see http://www.ncbi.nlm.nih.gov/SNP/; for CHOPCHOP, see http://chopchop.cbu.uib.no; for 1000 Genomes Project, see http://www.1000genomes.org/; for NHLBI ESP6500 databases, see http://evs.gs.washington.edu/EVS/; for VariantDB, see http://www.biomina.be/app/variantdb; for ANNOVAR, see http://www.openbioinformatics.org/annovar/; for NCBI Map Viewer, see http://www.ncbi.nlm.nih.gov/mapview/; for Exome Aggregation Consortium (ExAC), Cambridge, MA, see http://exac.broadinstitute.org; for Genome Aggregation Database (gnomAD), Cambridge, MA, see http://gnomad.broadinstitute.org.

## Data availability

The data that support the findings of this study are available from the corresponding author upon reasonable request.

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

## Acknowledgements

We thank the families for their participation and collaboration. We thank Tiziana Cogliati (NEI/NIH) for critical review of the manuscript; Drs. G. Walz and Michelle Dawn (University Freiburg Medical Center) for zebrafish *fat1* constructs; Dr. S. Miller and Omar Memon (NEI/NIH) for RPE cells. We thank Dr. Robert Farris (NEI, Biological Imaging Core) for expert assistance with confocal laser scanning microscopy, and Chris Ardeljan and Heba Mohammed (NEI/NIH) for technical assistance with electron microscopy. Andrew Wegerski and Tyler Fayard (NEI/NIH) for zebrafish breeding and maintenance. We thank Dr. Ahron Bloch, Prof. Michal Elhalel-Dranitzky, Prof. Dvora Rubinger, Dr. Amir Haze, Dr. Samer Khateb (Hadassah-Hebrew University Hospital, Jerusalem, Israel), and Prof. Francesco Sema-raro (Eye Clinic, Department of Neurological and Vision Sciences, University of Brescia, Italy) for their clinical assistance. We thank Leander Beekman for technical assistance (Academic Medical Center, Amsterdam). This work was supported by the Agence Nationale de la Recherche (ANR-10-IAHU-01), France and intramural program of the National Eye Institute/National Institutes of Health, USA (DK076683) to F.H. F.H. is the William E. Harmon Professor.

## Author contributions

C.R.B., B.P.P., and A.S. designed and supervised the study. N.L. and A.G. compiled results, wrote the manuscript, and prepared all the figures. N.L., I.R., A.H.-C., F.-Z.L., E.M.L., H.K., V.M., and S.M. performed sequencing data analysis. I.R., N.L., R.S., S.C.E., N.A., Y.L., H.E., I.C., V.L., A.P., J.A., S.L., B.W., A.B., and F.H. recruited patients and gathered detailed clinical information for the study. H.M. provided mouse strains and molecular biology reagents. S.B., K.B., F.O., and A.G. performed mouse studies. A.V.P. performed LOD score calculation and critically reviewed the manuscript. K.B. supervised and provided cell lines, reagents and funds for cell culture experiments. A.G. and R.S. performed cell culture experiments. A.G. performed zebrafish experiments. M.A. performed electron microscopy studies. All the authors read and agreed on the final draft of the manuscript.

## Additional information

**Competing interests:** The authors declare no competing interests.

Najim Lahrouchi[1] Aman George[2], Ilham Ratbi[3], Ronen Schneider[4], Siham C. Elalaoui[3], Shahida Moosa[5,6], Sanita Bharti[2,7], Ruchi Sharma[7], Mones Abu-Asab[8], Felix Onojafe[2], Najlae Adadi[3], Elisabeth M. Lodder [1], Fatima-Zahra Laarabi[9], Yassine Lamsyah[10], Hamza Elorch[10], Imane Chebbar[10], Alex V. Postma [11], Vassilios Lougaris[12], Alessandro Plebani[12], Janine Altmueller[13,14,15], Henriette Kyrieleis[16], Vardiella Meiner[17], Helen McNeill[18], Kapil Bharti[7], Stanislas Lyonnet[19], Bernd Wollnik[5], Alexandra Henrion-Caude[20], Amina Berraho[10], Friedhelm Hildebrandt [4], Connie R. Bezzina[1], Brian P. Brooks[2] & Abdelaziz Sefiani[3,9]

[1]Amsterdam UMC, University of Amsterdam, Heart Center, Department of Clinical and Experimental Cardiology, Amsterdam Cardiovascular Sciences, Meibergdreef 9, Amsterdam 1105AZ, The Netherlands. [2]Ophthalmic Genetics and Visual Function Branch, National Eye Institute, NIH, Bethesda, MD 20892, USA. [3]Centre de Recherche en Génomique des Pathologies Humaines (GENOPATH), Faculté de Médecine et de Pharmacie, Mohammed V University of Rabat, 10100 Rabat, Morocco. [4]Department of Pediatrics, Boston Children's Hospital, Harvard Medical School, Boston, MA 02115, USA. [5]Institute of Human Genetics, University Medical Center Goettingen, 37073 Goettingen, Germany. [6]Boston Children's Hospital and Harvard Medical School, Boston, MA 02215, USA. [7]Unit on Ocular & Stem Cell Translational Research, National Eye Institute, NIH, Bethesda, MD 20892, USA. [8]Section of Histopathology, National Eye Institute, NIH, Bethesda, MD 20892, USA. [9]Département de génétique médicale, Institut National d'Hygiène, BP 769 Agdal, 10090 Rabat, Morocco. [10]Service d'Ophtalmologie B, Hôpital des Spécialités, CHU Rabat, Faculté de médecine et de Pharmacie, Mohammed V University of Rabat, 10100 Rabat, Morocco. [11]Amsterdam UMC, University of Amsterdam, Department of Anatomy, Embryology & Physiology, Amsterdam Cardiovascular Sciences, Meibergdreef 9, Amsterdam, The Netherlands. [12]Pediatrics Clinic and Institute for Molecular Medicine "A. Nocivelli", Department of Clinical and Experimental Sciences, University of Brescia and ASST-Spedali Civili of Brescia, 25123 Brescia, Italy. [13]Cologne Center for Genomics University of Cologne, 50931 Cologne, Germany. [14]Center for Molecular Medicine Cologne (CMMC), University of Cologne, Cologne 50931, Germany. [15]Institute of Human Genetics, University of Cologne, 50931 Cologne, Germany. [16]Department of Pediatrics, Bethanien Hospital, Cologne 42699, Germany. [17]Department of Human Genetics and Metabolic Diseases, Hadassah-Hebrew University Medical Center, Jerusalem 91120, Israel. [18]Department of Developmental Biology, Washington University School of Medicine, St. Louis 63110 MO, USA. [19]Laboratory of embryology and genetics of human malformation, Institut National de la Santé et de la Recherche Médicale (INSERM) UMR 1163, Paris Descartes-Sorbonne Paris Cité University, Institut Imagine, Paris 75015, France. [20]INSERM UMR-781, Département de Génétique, Hôpital Necker-Enfants Malades, Assistance Publique Hôpitaux de Paris (APHP), Paris 75015, France. These authors contributed equally: Najim Lahrouchi, Aman George, Ilham Ratbi. These authors jointly supervised the work: Connie R. Bezzina, Brian P. Brooks, Abdelaziz Sefiani.

