## [Peer Review File · Nature Communications]

Reviewer #1 (Remarks to the Author):

Referee comments on NCOMMS-18-06101-T.

In the manuscript entitled “Truncating mutations in the atypical protocadherin 1 FAT1 cause a recessive clinical syndrome characterized by colobomatous microphthalmia, blepharoptosis, nephropathy and syndactyly” Lahrouchi, Ratbi, George and colleagues present novel findings resulting from mutations in the FAT1 gene.

They identified five families presenting a “novel syndrome” resulting from different mutations of the FAT1 gene. The syndrome was involving pathologies like facial dysmorphism, syndactyly, nephropathy, and several pathologies affecting the eye, like blepharoptosis, microphthalmia and coloboma. The latter showed high penetrance and was the major focus of the further analyses of the manuscript. The authors consecutively made use of Knock Out mice, zebrafish morphants and zebrafish mutants to address the coloboma phenotype in more detail. This part of the study is highly attractive. In order to provide cell biological insights, the authors used a cell culture model of human RPE cells.

In general I find the fact very appealing that the authors addressed the coloboma phenotype further using next to the identified individuals from the affected families also KO mice and zebrafish. Although the coloboma phenotype per se seems clear to me I find the conclusion made from the morphological findings potentially incorrect. The authors linked the phenotype they identified to a defect in optic fissure fusion. A failure during the process of optic fissure fusion is probably the most commonly appreciated cause for coloboma formation. The morphological data presented, however, could be pointing towards a morphogenetic defect during optic fissure morphogenesis, which consecutively leads to coloboma formation. The cell biological findings provided would be fitting to a morphogenetic defect as well. I want to stress the point that this would be highly interesting also for the scientific community.

Specific points:

The images presented showing the blepharoptosis and the syndactyly in Figure 1 are clear. The kidney involvement could have also been documented, however the effect of an FAT1 mutation on the kidney has already been described. The presentation of the colobomatous microphthalmia would have benefitted from an analysis with an optical coherence tomography. Here especially the morphology of the persisting fissure margins could give further insights. The aspect of microphthalmia was raised. In Table 1 the result of the assessment of the patients are given. How was the size of the human eyes measured?

In Figure 2 images of in situ hybridizations are presented. The axis for orientation seems to be fitting for a-c but not for d-f. In g-o FAT KO eyes and wildtype eyes are compared. I appreciate the presentation of the phenotype. Are k and n taken at a comparable “height/ depth”. I assume that

the these are sagittal sections? Considering the morphology of the fissure margins, it seems as if these are not properly formed. The presumptive neuroretinal domain is everted and localized in the lens-averted domain. This could be indicative of a morphogenetic defect. Thus, also a more precise analysis of the fissure margins of the human eyes would be helpful. Of course, secondary effects could also alter the morphology of the eye over time.

In Figure 3 the zebrafish morphant data is presented. In Figure 3 b and c the presentation of the eyes would have benefitted from a different illumination. Illumination from bottom and top could help here. I appreciate that also the whole embryo is shown in b and c. In d-g histological sections are presented. I assume that these are sagittal sections. Please specify the section angle. The morphology of the coloboma phenotype is different between c and f/g. I assume that c, f and g also show 3 days pf embryos? What would be the explanation for the huge differences in between the phenotypes presented? I would like to see histological sections from an embryo showing the phenotype in c. How do the fissure margins look like? In h and I the expression of fat1a is addressed. Here I would like to see more details, especially with respect to the expression domains within and around the eye. In addition, the expression of fat1b is of interest, even though, a coloboma is visible in the fat1a single morphant and a total redundancy between fat1a and fat1b cannot be expected. I see the point that a rescue experiment is difficult to perform based on the size of the fat1a mRNA. Furthermore, mRNA based overexpression of fat1a will likely also have detrimental effects in itself.

From my point of view the supplement Figure 4 could/ should in part be integrated in the Figure 3. In supplement Figure 4, zebrafish mutants for fat1a are addressed. CRISPR/Cas9 was used to induce a site directed mutagenesis. It is mentioned that mouse Fat1 and Fat4 are similar in the extracellular domains, however, differ in the intracellular C- terminal domains. Since Fat1 mutants show a coloboma and Fat4 mutant do not, the rationale for the site directed genome editing was to truncate the C-terminal region (ENA/VASP and PDZ domains). I appreciate this step, which is further specifying the important domains of fat1a for fissure closure. However, it is also mentioned in line 171 (page7) that “this” mimics the genetic variants found in the affected individuals. This I cannot follow, since the data presented in Figure 1b is showing that in families 1, 2, 3 and 5 the mutations were found closer to the N-terminal region. Please specify what was meant here. In the short technical description, given for the CRISPR experiments in the materials and methods section, it is stated that mRNA was generated from cDNA. Was this then used as sgRNA? This seems not to be a standard approach for CRISPR/Cas9 experiments. Please give a more precise description of the technique including vectors which were used and of databases which were used to predict suitable PAM sites. Sequencing reads are presented to show the small “indel” in the fat1a gene. A truncation of the C-terminal domains was thus achieved. Importantly the phenotype resulting, suggests a morphogenetic defect. I would like to see a more detailed presentation of the phenotype, showing the morphology of the fissure margins. Histology could help here.

In the next part, in Figure 4 and 5, the authors address the cell biological function of Fat1. Based on the coloboma the authors identified, they assume that Fat1 plays an important role during optic fissure fusion. As mentioned above, a failure during the process of optic fissure fusion is probably the most commonly appreciated cause for coloboma formation. However, also an affected morphogenesis of the optic cup and optic fissure results in coloboma. The authors furthermore mention that a the RPE cells are at the leading edge of the fissure margins that mediate the closure. Thus, they use RPE cells for their further analyses. At this point I would like to question whether it is

exactly clear which “type of cell” is initiating the fusion. Please specify what data the statement is based on.

What was the rationale to use RPE cells isolated from human cadavers. Is the behavior of old cells comparable to that of embryonic cells? I appreciate the effort to get as close as possible to the human context. However, in this point I think that mouse embryonic RPE cells would have been suited better.

In Figure 4 a colocalization of FAT1 with F-actin and tight junctions is addressed at 48h (a) after seeding and after 8 weeks in culture (b). In Figure 4a I appreciate the localization of Fat1 at sites where also domains of F-actin are localized. What exactly is this domain corresponding to? To which of the cells does the FAT1 stained domain belong to? I also appreciate the localization at “forming” tight junctions. Though it is difficult to interpret the arrangement of the ZO-1 staining in itself. Could the membranes be made visible? Is the cell membrane folded? The authors state that these are the earliest cell-cell contacts. Please specify how this was shown.

In Figure 4b Fat1 seems to be localized at the cell membranes, where also a terminal web of actin filaments can be found. With respect to a colocalization of FAT1 with tight junctions I cannot follow. A close up of a cell with a “yellowish” membrane is shown. In most of the other cells, however, only a green label can be seen. How were the images acquired? Could the differential “colocalization” be a result from a scan of different cellular heights? It was mentioned that super resolution confocal imaging was used. Could this be specified? Are single optical planes presented? Overall the FAT1 stainings seem to potentially have background. Please show also the controls of FAT1 immunos on FAT1 KO tissue, which were mentioned in the text.

In Figure 5 an shRNA mediated knockdown of FAT1 in RPE cells is analyzed. I appreciate the presentation of the western blots and the densitometric quantification showing the intensity of the knockdown, which is approximately 60%. I appreciate that the “persistence/existence” of tight junctions after Fat1 knockdown was addressed also with electron microscopy. In Figure 5b, however, I find it difficult to follow. I think that there might still be a tight junction in the junctional domain the upper white arrow is pointing to in the lower image. On the other hand, I find it difficult to see a tight junction at the site where the upper white arrow is pointing to in the upper image. An image with higher resolution will probably help. Overall it is crucial to know the orientation in which the sections were made. Are these tangential sections or do we see the cells sitting next to each other from the side? In Figure 5c the changes in F-actin composition resulting from a Fat1 knockdown are addressed. As mentioned above, the immunohistochemistry seems to have a background. The overall staining of Fat1 seems not reduced by 60%. I appreciate, however, that the labeling of the membranes is largely absent after knockdown. In addition the knockdown seems to affect the F-actin which is associated with the cell membranes. In Figure 5d the effect on tight junctions is addressed. Here the overall staining of ZO-1 seems to differ largely to the homogeneous staining achieved in Figure 4. What could be the reason for this? Here the Fat1 immuno seems to have less background. The knockdown of ZO-1 seems to be 100% according to the image presented. Seeing also potential general problems with the ZO-1 immuno presented in Figure 5, I would suggest to repeat this experiment to be on the safe side. The effect measured with the test of the resistance seems to be strong. Could statistics be performed also between the control group and the knockdown group? Is the test for the resistance also a good measure for the physical integrity of the epithelium?

Overall I think that the data presented do not allow the conclusion drawn in line 151 (page6).

As pointed out, I rather think that the morphological data presented could be pointing towards a morphogenetic defect during optic fissure formation, which consecutively results in coloboma. The authors should address the morphology of the fissure margins in the different conditions (human, FAT1 KO mouse, zebrafish mutant) more precisely and then decide whether a morphogenetic defect is more likely than a hampered fusion. Other possible functions of Fat1, which are known from the literature and mentioned in the introduction would also be fitting to the observations made.

Reviewer #2 (Remarks to the Author):

Human Data

This paper reports homozygous loss-of-function mutations in *FAT1* associated with a syndromic condition characterised by reasonably penetrant optic fissure closure defect and unusual soft tissue or osseous 3/4 syndactyly. A less penetrant but distinctive feature is nephropathy. Four ultra rare LOF alleles were identified in five apparently unrelated families. The two families who shared an allele are from the same country. Generally the human genetics and clinical phenotype are well presented

Minor points

A combined LOD (not lod as in text) score for the locus should be calculated using all families

YAP1 is another human locus which is associated with ocular coloboma and renal dysfunction albeit non-progressive - I was surprised this was not mentioned given the plausible functional interaction of these two gene products in the HIPPO pathway

Failure of epithelial fusion is not the only mechanisms for coloboma, failure of growth in the vesicle can result in failure of the optic fissure to appose

Mouse Data

The null mouse embryos seem to have a more severe phenotype than the human probands. However non closure of the optic fissure is apparent in the embryos. I could not see any mention of syndactyly in the 14.5 dpc mouse embryo.

Minor points

Mouse RNA should be designated *Fat1* rather than *FAT1* as far as I am aware - if this is correct then the manuscript and figure legends should be carefully checked for consistent usage.

Zebrafish

I am not really competent to comment in detail but there are clearly issues with interpretation of morphants in paralogs - zebrafish embryos have a reputation of developing coloboma very easily

Reviewer #3 (Remarks to the Author):

In this paper FAT1 is proposed to be at the heart of optic fissure closure. Four different FAT1 protein-truncating mutations in five unrelated families with a complex syndrome convincingly show the cause of disease. By combining immunolocalization, mouse and zebrafish model studies, the authors make a strong case for the mechanism of disease and function of FAT1. The paper is very well written, the results are displayed in a clear manner and I found no major technical omissions.

The fact that protein-truncating mutations result in a complex syndrome and that missense variants merely result in nephropathy is of interest. It is surprising that not all cases in this report show nephropathy. The authors should further discuss this.

Minor comments:

In legends and the table, the protein mutation notations are not uniform; some in one, some in 3-letter code. Make uniform in one-letter code.

line 207: c.9093....; contains an error. Probably must be c.3093..., with protein notation: p.P1032Cfs*11.

Not all legends of figures contain a general title that captures the content of these figures. Please insert.

Reviewer # 1 (remarks to the Author)

We thank the reviewer for his/her comments. As the original version of the manuscript had been written in a letter format (transferred to Nature Communications from Nature Genetics to which we had originally submitted) our representation of the clinical and the animal model phenotype and the discussion of the findings had been limited due to space constraints. This has now been adjusted, taking into account the reviewers' suggestions. Please find our point by point response below:

The images presented showing the blepharoptosis and the syndactyly in Figure 1 are clear. The kidney involvement could have also been documented, however the effect of an FAT1 mutation on the kidney has already been described.

The reviewer is correct; kidney involvement was not discussed in detail as in the initial concise letter format we opted to focus primarily on the optic fissure closure defects. However, we agree with the reviewer that a detailed description of the patient phenotype is warranted. We now added renal histology and electron microscopy figures in **Supplementary Figure 2**. In addition, the clinical findings of each patient including renal phenotype are described in **Table 1** of the manuscript. We also added a section to the Discussion dedicated to kidney involvement in our patients.

The presentation of the colobomatous microphthalmia would have benefitted from an analysis with an optical coherence tomography. Here especially the morphology of the persisting fissure margins could give further insights.

We thank the reviewer for the suggestion. The OCT images of patients are now included in **Supplementary Figure 1** and the text has been modified accordingly.

The aspect of microphthalmia was raised. In Table 1 the result of the assessment of the patients are given. How was the size of the human eyes measured?

The size of the eye was determined by measuring the axial length of the eye with an echo-biometer. We have added this information to the manuscript.

In Figure 2 images of in situ hybridizations are presented. The axis for orientation seems to be fitting for a-c but not for d-f.

The axis representing orientation has been replaced with notations depicting the nasal and temporal regions in the sagittal section and dorsal and ventral regions in the coronal sections.

In g-o FAT KO eyes and wildtype eyes are compared. I appreciate the presentation of the phenotype. Are k and n taken at a comparable "height/ depth"

The images "k and n" were taken at similar magnification (10X). Though we do agree with the reviewer that it does seem to be at different depths. We think this could be due to the original size of the mouse embryo, which in some instances we have observed to be variable especially for the knockout embryos.

I assume that these are sagittal sections?

Images “h-o” are sagittal sections and this information has now been incorporated in the legend of **Figure 2**.

Considering the morphology of the fissure margins, it seems as if these are not properly formed. The presumptive neuroretinal domain is everted and localized in the lens-averted domain. This could be indicative of a morphogenetic defect. Thus, also a more precise analysis of the fissure margins of the human eyes would be helpful. Of course, secondary effects could also alter the morphology of the eye over time.

We agree with the reviewer’s suggestion that coloboma due to loss of FAT1 could be due to morphogenetic defects as well. For this very reason, we choose to depict the phenotype (**Figure 2 j-l**) due to intervening peri-ocular mesenchyme. The reason we think that it is a RPE mediated defect, at least in part, since we observed strong mRNA staining in the presumptive RPE and weak in the presumptive neural retina. Also, we do not exclude the possibility of involvement of peri-ocular mesenchyme since FAT1 mRNA expression was observed there as well.

In Figure 3 the zebrafish morphant data is presented. In Figure 3 b and c the presentation of the eyes would have benefitted from a different illumination. Illumination from bottom and top could help here. I appreciate that also the whole embryo is shown in b and c.

The images were acquired again using the illumination settings suggested by the reviewer and **Figure 3** has been modified accordingly.

In d-g histological sections are presented. I assume that these are sagittal sections. Please specify the section angle.

The images “d-g” are sagittal sections and this information has now been incorporated in the legend of **Figure 3**. The section angle is also specified now by depicting the nasal and temporal fissure margins.

The morphology of the coloboma phenotype is different between c and f/g. I assume that c, f and g also show 3 days pf embryos? What would be the explanation for the huge differences in between the phenotypes presented? I would like to see histological sections from an embryo showing the phenotype in c. How do the fissure margins look like?

Morpholino injection usually generates a range of phenotypes varying from mild to severe and we have observed this to be true for morpholino-mediated knock-down of *fat1a* as well. Furthermore, our lab routinely characterizes coloboma as a consequence of morpholino-mediated knock-down in mild, moderate and severe categories. In the current manuscript, we have used two different concentrations of the morpholino and we usually observe all the three above-mentioned categories of coloboma at different concentration, with the severe phenotype being more prevalent at higher doses. But for the sake of this manuscript we decided to show data in the form of presence or absence of coloboma, so as to focus on the clinical phenotype observed and the consistency of the coloboma phenotype across species. We depicted a more severe form of coloboma, which would be easily appreciated by a wider readership, including individuals not

expert in the field. With regard to histology we have now chosen to represent the phenotype which was consistent with the mouse and CRISPR knockout zebrafish phenotype. We have modified **Figure 3b and c** images which are more in line with the **Figure 3f and g**. The original images are now moved to the **Supplementary Figure 4** along with the corresponding histology images as requested by the reviewer. As requested by the reviewer we have also included the histological pictures for phenotype depicted in original **Figure 3c**.

In h and I the expression of *fat1a* is addressed. Here I would like to see more details, especially with respect to the expression domains within and around the eye. In addition, the expression of *fat1b* is of interest, even though, a coloboma is visible in the *fat1a* single morphant and a total redundancy between *fat1a* and *fat1b* cannot be expected. I see the point that a rescue experiment is difficult to perform based on the size of the *fat1a* mRNA. Furthermore, mRNA-based overexpression of *fat1a* will likely also have detrimental effects in itself.

Higher magnification images for *fat1a in situ* staining are now provided with a focus on the expression pattern in and around the eye. It should be noted that *fat1a* and *fat1b* expression is not restricted to specific domains and is rather unrestricted and present at the rostral end (ZFIN) of the embryo and absent at the caudal end, during the time of optic fissure closure. The same is true for *fat1b* expression as well. Zebrafish *fat1a* and *fat1b in situ* staining are now represented in a new figure in the supplementary section (please refer to **Supplementary Figure 3**).

From my point of view the supplement Figure 4 could/ should in part be integrated in the Figure 3.

The reason we kept the morpholino and CRISPR data separate since the CRISPR experiment was performed with a very specific aim of deleting a domain rather than creating a general knockout of the *fat1a* gene. As such we thought it would easier to explain the reason behind this experiment if it was kept as an independent experiment rather than combining with the morpholino experiment which is the disruption the *fat1a* protein formation. **Supplementary figure 4** has now been incorporated into the main text as a new figure (**Figure 7**), as suggested by the reviewer.

In supplement Figure 4, zebrafish mutants for *fat1a* are addressed. CRISPR/Cas9 was used to induce a site directed mutagenesis. It is mentioned that mouse Fat1 and Fat4 are similar in the extracellular domains, however, differ in the intracellular C- terminal domains. Since Fat1 mutants show a coloboma and Fat4 mutant do not, the rationale for the site directed genome editing was to truncate the C-terminal region (ENA/VASP and PDZ domains). I appreciate this step, which is further specifying the important domains of *fat1a* for fissure closure. However, it is also mentioned in line 171 (page7) that “this” mimics the genetic variants found in the affected individuals. This I cannot follow, since the data presented in Figure 1b is showing that in families 1, 2, 3 and 5 the mutations were found closer to the N-terminal region. Please specify what was meant here.

We understand the concern of the reviewer regarding the nature of the homozygous truncating *fat1a* mutation in zebrafish, as such we have removed the sentence and replaced it with a more appropriate sentence describing the nature of the mutation in the fish model.

In the short technical description, given for the CRISPR experiments in the materials and methods section, it is stated that mRNA was generated from cDNA. Was this then used as sgRNA? This seems not to be a standard approach for CRISPR/Cas9 experiments. Please give a more precise description of the technique including vectors which were used and of databases which were used to predict suitable PAM sites.

We apologize for this oversight. The sentence “*Zebrafish fat1a cytoplasmic domain coding cDNA was used to in vitro synthesize mRNA (mMACHINE kit, Ambion, Grand Island, NY) for injections*” was incorporated there erroneously (editing mistake) and has been removed. This was not used as the sgRNA. A more detailed and correct description of the CRISPR/Cas9 genome editing technique we used has now been incorporated in the Methods section.

Sequencing reads are presented to show the small “indel” in the fat1a gene. A truncation of the C-terminal domains was thus achieved. Importantly the phenotype resulting, suggests a morphogenetic defect. I would like to see a more detailed presentation of the phenotype, showing the morphology of the fissure margins. Histology could help here.

As requested by the reviewer, detailed histology of the fat1a CRISPR edited fish has been included in the newly added **Figure 7**. To address the concern raised by the reviewer we now performed histology right before optic margin fusion to study the morphology of fissure margins, as requested by the reviewer. We observed defects in the organization of the optic cup, integrity of the retinal pigment epithelium and defective organization of optic fissure margins.

In the next part, in Figure 4 and 5, the authors address the cell biological function of Fat1. Based on the coloboma the authors identified, they assume that Fat1 plays an important role during optic fissure fusion. As mentioned above, a failure during the process of optic fissure fusion is probably the most commonly appreciated cause for coloboma formation. However, also an affected morphogenesis of the optic cup and optic fissure results in coloboma.

We very much thank the reviewer for this comment. Indeed, while a failure to fuse the fissure margins during the process of optic fissure fusion is a commonly appreciated cause for coloboma formation, it could also be due in part to morphogenetic events. Our assumption that FAT1 truncating variants cause coloboma as consequence of a primary fusion defect is based on the fact that FAT1 knockout mice exhibit neural tube closure defects and epithelial podocyte junctional defects. Also, based on previous studies, FAT1 seems to play an important role during earliest cell-cell adhesion in epithelial cells. It is plausible that the coloboma observed in Fat1^{-/-} mice is primarily an epithelial cell mediated defect. However, indeed, at this moment we cannot exclude the possibility of morphogenetic defects being another factor causing the coloboma phenotype. This can be appreciated in the panels of **Figure 2 (k and l panels)** where intervening peri-ocular mesenchyme can be easily observed obstructing fusion of fissure margins. We now comment on the issue raised by the reviewer in the Discussion section of the manuscript.

The authors furthermore mention that a the RPE cells are at the leading edge of the fissure margins that mediate the closure. Thus, they use RPE cells for their further analyses. At this point I would like to question whether it is exactly clear which “type of cell” is initiating the fusion. Please specify what data the statement is based on.

The classical ultra-structure studies on optic fissure fusion, performed by Isabella Hero describe RPE as the cell type that is the initial focus of fusion (**REF1**). The reference has been included in the manuscript. Though it should be noted that at this time point the presumptive RPE has not yet terminally differentiated into the classical pigmented RPE. This is important as it forms the basis of our later experiments where we culture RPE at sub-confluent density to mimic a mesenchymal state rather than the classical epithelial mono-layer.

REF1: Optic Fissure Closure in the Normal Cinnamon Mouse. *Investigative Ophthalmology & Visual Science*, Vol. 31, No. I, January 1990. Page 204 para2

What was the rationale to use RPE cells isolated from human cadavers. Is the behavior of old cells comparable to that of embryonic cells? I appreciate the effort to get as close as possible to the human context. However, in this point I think that mouse embryonic RPE cells would have been suited better

The major reason to use cultured primary RPE cells was to recapitulate the earliest events of the junction formation. As suggested by the reviewer the rationale for using human fetal RPE cells was to get as close as possible to the human context. However, we wish to point out that the RPE used were of fetal origin (16-22 weeks gestational age) and not adult; this has now been mentioned in the Methods section. During human gestation, the optic fissure closes around 6-7 weeks of gestation. The RPE cells were typically used within 1 or 2 passage number. Also, as requested by the reviewer we performed additional experiments and provide data using mouse RPE flat mounts isolated from E14.5 WT and *Fat1*^{-/-} mouse optic cup. We did not observe any drastic defects in F-Actin organization and ZO-1 staining patterns between the WT and *Fat1*^{-/-} mouse RPE. As such it does seem that FAT1 plays an important role in epithelial cell mediated fusion rather than differentiation of RPE from neuro-ectodermal lineage.

In Figure 4 a colocalization of FAT1 with F-actin and tight junctions is addressed at 48h (a) after seeding and after 8 weeks in culture (b). In Figure 4a I appreciate the localization of Fat1 at sites where also domains of F-actin are localized. What exactly is this domain corresponding to? To which of the cells does the FAT1 stained domain belong to

The exact nature of these domains is currently ambiguous in the literature, they are sometimes referred to as cytoplasmic bridges/processes or filopodial projections (**REF1**). The FAT1 staining belongs to RPE cells, though it is not possible to conclude from the current experiments as to exactly which cells, out the two interacting partners, the FAT1 staining belongs to. But from our studies in **Supplementary Figure 1b**, and also as reported by other authors (**REF 2 and 3**) FAT1 is present at the leading edges of the cells and tend to accumulate at the tips of filopodia and lamellipodia (**REF 2 and 3**).

REF1: Vasioukhin, V., Bauer, C., Yin, M. & Fuchs, E. Directed actin polymerization is the driving force for epithelial cell-cell adhesion. *Cell* **100**, 209–19 (2000).

REF2: Tanoue, T. & Takeichi, M. Mammalian Fat1 cadherin regulates actin dynamics and cell–cell contact. *J. Cell Biol.* **165**, 517–528 (2004)

REF3: Hou, R. & Sibinga, N. E. S. Atrophin Proteins Interact with the Fat1 Cadherin and Regulate Migration and Orientation in Vascular Smooth Muscle Cells. *J. Biol. Chem.* **284**, 6955–6965 (2009).

I also appreciate the localization at “forming” tight junctions. Though it is difficult to interpret the arrangement of the ZO-1 staining in itself. Could the membranes be made visible? Is the cell membrane folded? The authors state that these are the earliest cell-cell contacts. Please specify how this was shown.

The high magnification panels in **Figure 4a** are super-resolution images. Technically it is not possible to include a transmitted light photo multiplier tube (T-PMT) while acquiring super-resolution images with current imaging modalities, adding transmitted light would lead to loss of super-resolution abilities. As such the membranes cannot be made visible. The boundaries of neighboring cells are clearly demarcated in the low magnification image in the first panel. With the current staining pattern presented it is not possible to discern if the membrane is folded. ZO-1 has been shown to localize at earliest RPE cell-cell contacts previously (**REF1**), and as such the staining pattern of ZO-1 was taken as the reference point.

REF1: Economopoulou M, Hammer J, Wang F, Fariss R, Maminishkis A, Miller SS. Expression, localization, and function of junctional adhesion molecule-C (JAM-C) in human retinal pigment epithelium, *Invest Ophthalmol Vis Sci.* 2009 Mar;50(3):1454-63)

In Figure 4b Fat1 seems to be localized at the cell membranes, where also a terminal web of actin filaments can be found. With respect to a colocalization of FAT1 with tight junctions I cannot follow. A close up of a cell with a “yellowish” membrane is shown. In most of the other cells, however, only a green label can be seen. How were the images acquired? Could the differential “colocalization” be a result from a scan of different cellular heights? It was mentioned that super resolution confocal imaging was used. Could this be specified? Are single optical planes presented? Overall the FAT1 stainings seem to potentially have background. Please show also the controls of FAT1 immunos on FAT1 KO tissue, which were mentioned in the text.

The first panel (on the left) represents the merged maximum intensity projection (MIP) and NOT a single z-plane image. The image with FAT1 and ZO-1 staining has been replaced with a better image depicting a uniform staining pattern and colocalization. The images that were acquired by super-resolution mode are now mentioned in the figure legends. The F-Actin and ZO-1 staining (green) tend to be more specific and localized to the apical tight junctions only, as such they have a better signal/noise ratio. The FAT1 antibody is localized at the cell membrane and cytoplasm as a result the signal/noise ratio with this antibody is not good. Immunostaining for FAT1 in mouse embryonic fibroblast cells derived from WT and *Fat1*^{-/-} embryos is shown in **Supplementary Figure 5c**. This figure is now referred to in the text.

In Figure 5 an shRNA mediated knockdown of FAT1 in RPE cells is analyzed. I appreciate the presentation of the western blots and the densitometric quantification showing the intensity of the knockdown, which is approximately 60%. I appreciate that the “persistence/existence” of tight junctions after Fat1 knockdown was addressed also with electron microscopy. In Figure 5b, however, I find it difficult to follow. I think that there might still be a tight junction in the junctional domain the upper white arrow is point to in the lower image. On the other hand, I find it difficult to see a tight junction at the site where the upper white arrow is pointing to in the upper image. An image with higher resolution will probably help. Overall it is crucial to know

the orientation in which the sections were made. Are these tangential sections or do we see the cells sitting next to each other from the side?

The electron microscopy images are now provided in a separate figure (**Figure 6**). A low magnification image depicts the orientation of the RPE monolayer on top of the trans-well membrane, and a high magnification image focus on the apical tight junctions. These are transverse sections depicting two or more RPE cells sitting next to each other (**Figure 6a**). In FAT1 shRNA treated cells the RPE cells fail to organize in a monolayer (**Figure 6b**) and exhibit defective tight junction.

In Figure 5c the changes in F-actin composition resulting from a Fat1 knockdown are addressed. As mentioned above, the immunohistochemistry seems to have a background. The overall staining of Fat1 seems not reduced by 60%. I appreciate, however, that the labeling of the membranes is largely absent after knockdown. In addition the knockdown seems to affect the F-actin which is associated with the cell membranes.

As recommended by the reviewer a more appropriate image depicting FAT1 knockdown and disorganized F-Actin fibers is now provided (**Figure 5b**).

In Figure 5d the effect on tight junctions is addressed. Here the overall staining of ZO-1 seems to differ largely to the homogeneous staining achieved in Figure 4. What could be the reason for this?

The immuno-staining experiments described in **Figure 5** were performed on RPE cells 2 weeks post seeding. This is the earliest possible time point by which the RPE cells start to organize in a monolayer and start to attain polarized epithelial phenotype. Also, F-Actin fibers and ZO-1 proteins start to organize at the apical tight junctions by this time point. In contrast, the **Figure 4b** RPE were immuno-stained 8 weeks post seeding. By this time the RPE cells are completely arranged in a polarized epithelial monolayer with matured tight junctions. As such the immuno-staining looks more homogenous in **Figure 4b**.

Here the Fat1 immuno seems to have less background.

In **Figure 5d**, the images were acquired and presented at lower magnification as compared to **Figure 4**, which requires lower laser power and/or gain resulting in reduced signal to noise ratio. This could have been the probable reason for the differences in the staining pattern.

The knockdown of ZO-1 seems to be 100% according to the image presented. Seeing also potential general problems with the ZO-1 immuno presented in Figure 5, I would suggest to repeat this experiment to be on the safe side. The effect measured with the test of the resistance seems to be strong.

We have now performed this experiment multiple times and have obtained consistent results. Currently we do not know the reason behind the dramatic loss of ZO-1 staining. One possible explanation can be the rapid turnover or cytoplasmic clearance of the protein. As can be observed in the **Figure 4** and **Figure 5**, that the cytoplasmic staining for ZO-1 is minimal in the cytoplasm and it is accumulated only at the earliest cell-cell contact in sub-confluent cultures and

apical tight junction in confluent RPE monolayers. Suggesting it is rapidly degraded from the cytoplasm. We presume that since it has not been accumulated/targeted/recruited at the apical tight junctions in the absence of FAT1 it is being rapidly degraded. We suspected that FAT1 might have a role in recruiting the ZO-1 protein to the apical tight junctions. This assumption is further supported by the fact that FAT1 has a C-terminal PDZ domain binding domain (HTEV) and ZO-1 has multiple PZD domains. We repeated the experiments again as suggested by the reviewer, but this time we stained with β -CATENIN and ZO-1 instead of FAT1 and ZO-1 (**Supplementary Figure 7**). In the scrambled group the β -CATENIN staining, which is present at Adherens junctions can be seen at cell borders and ZO-1 staining, which should present at the apical tight junctions has started to accumulate (A more uniform staining patten is seen when the cultures are completely polarized which takes 8 week). On the contrary in FAT1 shRNA treated cells there is some disruption of Adherens junctions (disruption of β -CATENIN staining) but the ZO-1 is completely gone suggesting the apical tight junctions are severely affected by the lack of FAT1.

Could statistics be performed also between the control group and the knockdown group?

The statistics cannot be performed between the control and FAT1 shRNA treated groups as the starting trans-epithelial resistance (TER) between the two groups is different. Also, this will not truly depict the nature of the experiment. With time the junctions between the RPE cells mature as a result the TER increases and reaches a steady state around 4-8 weeks. With the TER experiment performed we want to show that with time FAT1 shRNA treated group do not exhibit maturation of junctions as such the TER does not increase significantly whereas the scrambled treated group the TER increases significantly with time.

Is the test for the resistance also a good measure for the physical integrity of the epithelium?

Under in vitro conditions the TER is a measure of the barrier function i.e. tight junctional integrity of the RPE cells. TER measurements are widely used as real-time, non-destructive, and label-free measurements of epithelial and endothelial barrier function (**REF1**). Compromised cell-cell tight junctions results in low TER. Initially when the RPE cells are seeded on the trans-well membrane there is no or negligible TER. Over the course of 8 weeks, TER eventually increases as a result of maturation of junctions between the RPE cells and reaches steady state level.

REF1: Odijk, M. *et al.* Measuring direct current trans-epithelial electrical resistance in organ-on-a-chip microsystems. *Lab Chip* **15**, 745–52 (2015).

Reviewer # 2 (Remarks to the Author)

We thank the reviewer for his/her comments. Please find our answers below:

A combined LOD (not lod as in text) score for the locus should be calculated using all families.

We have now calculated a combined LOD score using all families and added the methods and results to the manuscript. We also changed “lod” to LOD in the text.

YAP1 is another human locus which is associated with ocular coloboma and renal dysfunction albeit non-progressive - I was surprised this was not mentioned given the plausible functional interaction of these two gene products in the HIPPO pathway.

This is indeed an aspect that deserves to be mentioned. As the original version of the manuscript had been written in a letter format (transferred to Nature Communications from Nature Genetics to which we had originally submitted) we had limited space for discussion. We now dedicate a paragraph in the Discussion section to the role of YAP1 and FAT1 in Hippo signaling as well as its potential role in coloboma genesis

Failure of epithelial fusion is not the only mechanisms for coloboma, failure of growth in the vesicle can result in failure of the optic fissure to appose.

We very much thank the reviewer for this comment. Indeed, while a failure to fuse the fissure margins during the process of optic fissure fusion is a commonly appreciated cause for coloboma formation, it could also be due in part to morphogenetic events. Our assumption that FAT1 truncating variants cause coloboma as consequence of a primary fusion defect is based on the fact that FAT1 knockout mice exhibit neural tube closure defects and epithelial podocyte junctional defects. Also, based on previous studies, FAT1 seems to play an important role during earliest cell-cell adhesion in epithelial cells. It is plausible that the coloboma observed in *Fat1*^{-/-} mice is primarily an epithelial cell mediated defect. However, indeed, at this moment we cannot exclude the possibility of morphogenetic defects being another factor causing the coloboma phenotype. This can be appreciated in the panels of **Figure 2 (k and l panels)** where intervening peri-ocular mesenchyme can be easily observed obstructing fusion of fissure margins. We now comment on the issue raised by the reviewer in the discussion section of the manuscript.

The null mouse embryos seem to have a more severe phenotype than the human probands. However non closure of the optic fissure is apparent in the embryos. I could not see any mention of syndactyly in the 14.5 dpc mouse embryo.

We checked *Fat1*^{-/-} mouse embryos (E14.5-15.5, n=5) but we did not observe any gross morphological limb defect as seen in our patients. We do acknowledge that at these developmental time points the limb development might not be completed which could be a reason why limb defects were not observed. Another reason could be the background strain, which is particularly important as, for example, neural tube defects are observed only in certain strains (**REF1**) but not in others suggesting contribution of the genetic background to the ultimate observed phenotype. *Fat1*^{-/-} mouse are embryonically lethal and therefore it is difficult to get later developmental stages suitable for detection of limb defects.

REF1: Badouel, C. *et al.* Fat1 interacts with Fat4 to regulate neural tube closure, neural progenitor proliferation and apical constriction during mouse brain development. *Development* **142**, 2781–2791 (2015).

Mouse RNA should be designated Fat1 rather than FAT1 as far as I am aware - if this is correct then the manuscript and figure legends should be carefully checked for consistent usage.

We now designate mouse RNA as Fat1 throughout the manuscript and figure legends.

I am not really competent to comment in detail but there are clearly issues with interpretation of morphants in paralogs - zebrafish embryos have a reputation of developing coloboma very easily.

To ensure best practice in our zebrafish studies we have followed the guidelines laid down for morpholino experiments by Stainier *et. al.*, (Plos Genetics, 2017):

1. We have used two different morpholinos for validation, this was not previously mentioned, but this information has now been added to the methods section.
2. We used a standard control morpholino as an injection control, this information has now been added to the methods section.
3. We have used a morpholino which has been previously validated and well characterized in two published studies (Skouloudaki *et al.*, 2009 and Gee *et al.*, 2016) according to the guidelines provided by Stainier *et. al.*, 2017.
4. We performed morpholino experiments at two different concentrations followed by detailed histological analyses; this information has now been incorporated as **Supplementary Figure 2** as requested by the reviewer.
5. Crucially, the genotype-phenotype correlations that we find in the zebrafish studies are further substantiated by mouse model and human subjects.

Unfortunately given the size of *fat1a* mRNA (>15kb) we cannot perform the rescue experiments.

Importantly, we generated zebrafish embryos homozygous for truncating *fat1a* mutations using CRISPR/Cas9 and validated the morpholino results (**Figure 7** and **Supplementary Figure 9**).

Reviewer #3 (Remarks to the Author):

We thank the reviewer for his/her comments. Please find our answers below:

The fact that protein-truncating mutations result in a complex syndrome and that missense variants merely result in nephropathy is of interest. It is surprising that not all cases in this report show nephropathy. The authors should further discuss this.

We agree with the reviewer that the penetrance of nephropathy should be discussed in more detail and we have now added this to the Discussion section of the manuscript. We carefully assessed kidney function and the presence of proteinuria in all patients and identified asymptomatic proteinuria in two siblings from family 1 (patients IV-1 and IV-5). The full overview of observed phenotypes, including renal manifestations, is summarized in **Table 1** of the manuscript. In total 5 out of 10 patients had proteinuria; 2 had presented with severe kidney disease (F3-IV-1, F5-II-1) and 3 were asymptomatic. Of importance for clinical follow-up, patient F3-IV-3, developed intermittent proteinuria with normal kidney function at the age of 20 years. We added these findings to the Results and Discussion section of the manuscript. We also added renal biopsy images of patient F5-II-1 that displayed glomerulotubular nephropathy in Supplementary Fig. 2

In legends and the table, the protein mutation notations are not uniform; some in one, some in 3-letter code. Make uniform in one-letter code.

We adjusted the protein mutation notations in the legend and table using a one-letter code as suggested by the reviewer.

line 207: c.9093....; contains an error. Probably must be c.3093..., with protein notation: p.P1032Cfs*11.

This is indeed an error, which we have now corrected. It should have stated c.3093_3096del.

Not all legends of figures contain a general title that captures the content of these figures. Please insert.

We have now added a general title to the legend of all figures.

Reviewer #1 (Remarks to the Author):

Referee comments on Review of NCOMMS-18-06101A.

In this manuscript entitled “Truncating mutations in the atypical protocadherin FAT1 cause a recessive clinical syndrome characterized by colobomatous microphthalmia, blepharoptosis, nephropathy and syndactyly” Lahrouchi, Ratbi, George and colleagues present their revised work of a previous version of this manuscript.

The point by point rebuttal letter was well appreciated and I want to point out that many aspects were carefully addressed. However, I still have some open points.

- In figure 2j-l it seems as if the margins were not properly formed! This is why I pointed out a potential morphogenetic defect before.
- In figure 3 (old version c) a different phenotype was presented, not well fitting to the sections provided in f,g. The new pictures fit much better. However, the variability of the phenotype should be addressed.
- It was mentioned that morpholino based knock down can result in mild, moderate and severe phenotypes. Would the authors still claim that in all distinct categories the same pathogenesis is the basis? I appreciate that the other images are added to the supplemental figures.
- How consistent is the CRISPR phenotype? Especially in f the image suggests a morphogenetic defect. In the sections the RPE cells did not get into the fissure! At least the differentiation of these cells seems to lack behind.
- The argument with respect to the “first tight junctions” seems not plausible to me using anti-ZO1 antibodies. Would it not be more important to address this issue over time?
- Changing the image in 4b is not solving the question/ problem from before. What was the reason for the green membranes?
- The inclusion of the control images for the antibody is much appreciated. The background of the antibody is now clearly visible. Don’t the authors share the opinion that such a background could be a problem?
- The additional information and the new figure 6 is much appreciated. Besides defective junctional complexes the failure to assemble a monolayer could also be part of the phenotype.
- I want to thank the authors for explaining the trans-epithelial resistance measurement and the reason for the display of the data. This seems plausible.
- The OCT images are very much appreciated. Could maybe also a normal OCT image be added? Could perhaps be pointed out, whether the retinal differentiation in the neighboring region was affected? In b the retinal layering seems to be affected. I think that additional “focal planes” would be helpful.

Reviewer #2 (Remarks to the Author):

This is a reasonably straightforward human genetics report. It documents the discovery of homozygous likely loss of function mutations in the gene encoding FAT1 in five apparently unrelated families. Mutations in FAT1 have been previously implicated in a genetic renal disorder but the phenotype reported here is an interesting and novel multisystem disorder which will be of interest to clinical geneticists, ophthalmologists and paediatricians in particular. The human and clinical genetics data is reasonably well described (see comments below) and the mouse and zebrafish modelling are well presented and supportive of a critical role for this gene in eye development. The cell biology is less interesting but reasonable to include.

My only major problem with this manuscript is that it almost completely ignores what seems to me to be the most obvious and interesting function on FAT1 in this context – both molecularly and clinically. FAT1 inhibits YAP1 via phosphorylation (for review see PMID:23076869) and function loss of FAT1 has been shown to cause activation of YAP1 in a very nice recent Nature Communications paper (Nat Commun. 2018 Jul 9;9(1):2372. doi: 10.1038/s41467-018-04590-1). Given the remarkable similarity in the resulting clinical phenotypes I would suggest that looking for loss of YAP1 phosphorylation in the animal models would be a critical experiment for this paper

Minor points

Abstract;

Truncating is misleading and should be replaced with “null” or “likely null”

Suggest ptosis, which is widely used in English, as a replacement throughout the paper for blepharoptosis (which is a correct but slightly arcane term)

“developmental ophthalmic defect” is redundant in the

Suggest replacing “our patients” with “affected individuals” as possessive terms are inappropriate in clinical reports

Suggest replacing “placing it exquisitely at the precise cell type and location for” with “consistent with a role in”

Introduction

It is probably more accurate to say that the eye develops from the eye field within the neural plate rather than from the neural tube

Methods

The different sequencing protocols that are used for each family would be best presented as a table which also includes the mean coverage and numbers of variants called across the samples sequenced by each pipeline.

Reviewer #3 (Remarks to the Author):

My points were well addressed by the authors

Reviewer # 1 (remarks to the Author)

In figure 2j-1 it seems as if the margins were not properly formed! This is why I pointed out a potential morphogenetic defect before.

As pointed out by the reviewer in **Figure 2j-1** the margins seem indeed not properly formed and point to a potential morphogenetic defect. This is now clearly discussed in page 14 of the manuscript.

In figure 3 (old version c) a different phenotype was presented, not well fitting to the sections provided in f,g. The new pictures fit much better. However, the variability of the phenotype should be addressed. It was mentioned that morpholino based knock down can result in mild, moderate and severe phenotypes. Would the authors still claim that in all distinct categories the same pathogenesis is the basis? I appreciate that the other images are added to the supplemental figures.

The variability of the morpholino phenotype is addressed in **Supplementary Figure 4**. Since it is well known that morpholino mediated knockdown results in a variable phenotype (ranging from mild to severe) we used 2 different morpholino concentrations and observed a mild phenotype at low concentration and severe defects were noted at high morpholino concentration. Whether this variability is due to a morpholino dosage effect or other effects is currently unknown. Notwithstanding, the observations we made in the Fat1 morpholino knock down zebrafish were confirmed in the other two models that we studied, namely CRISPR FAT1 mutant zebrafish and in Fat1^{-/-} mouse embryos.

How consistent is the CRISPR phenotype? Especially in f the image suggests a morphogenetic defect. In the sections the RPE cells did not get into the fissure! At least the differentiation of these cells seems to lack behind.

The CRISPR phenotype in zebrafish is completely penetrant and highly consistent, in line with what we observed in Fat1^{-/-} mouse embryos where each homozygous embryo consistently displayed coloboma. As we have observed in Fat1^{-/-} mouse embryos, the presence of morphogenetic defects in CRISPR zebrafish mutants cannot be excluded and we do agree with the reviewer that the differentiation of RPE cells does seem to lag behind in the ventral eye region as compared to the RPE present in the dorsal region of the eye cup. This information has now been incorporated in the Results section of the manuscript (page 18).

The argument with respect to the “first tight junctions” seems not plausible to me using anti-ZO1 antibodies. Would it not be more important to address this issue over time?

The reviewer is right in that time-course studies are necessary to make such a point. We have now reworded this part of the text and we refer to the loss of the ZO-1 staining pattern as loss and/or disruption of junctional markers rather than “first tight junctions”.

Changing the image in 4b is not solving the question/ problem from before. What was the reason for the green membranes?

The F-Actin and ZO-1 staining (which are green) tend to be more specific and localized to the apical tight junctions only, as such they have a better signal/noise ratio. The FAT1 antibody is localized at the cell membrane and cytoplasm and as a result the signal/noise ratio with this antibody is not optimal.

The inclusion of the control images for the antibody is much appreciated. The background of the antibody is now clearly visible. Don't the authors share the opinion that such a background could be a problem?

During the first revision, we repeated all FAT1 localization staining experiments using a monoclonal antibody (Anti-Procadherin FAT1 Antibody, clone C257 MSDS, Millipore) rather than the polyclonal antibody that we had used previously and we observed a similar staining pattern. In addition, there was no improvement in signal to noise ratio and background staining with the new antibody. We do share the opinion that background noise is visible even with this new antibody. However, the fact that similar staining patterns were observed with different FAT1 antibodies in our study and in the literature (**REF1 and 2**), and the fact that membrane/junctional localization of FAT1 is well established in epithelial cells (**REF1**), gives us confidence that this data is trustworthy. Previous reports have also suggested a nuclear and cytoplasmic staining pattern for FAT1 due to the cleavage of the full-length FAT1 protein (**REF1 and 3**). The antibody that we have used in our study is directed against the C-terminal end of the FAT1 protein further raising the possibility of detecting the FAT1 full length protein and its cleavage products.

REF1: Tanoue, T. & Takeichi, M. Mammalian Fat1 cadherin regulates actin dynamics and cell-cell contact. *J. Cell Biol.* **165**, 517–528 (2004).

REF2: Sadeqzadeh E, de Bock CE, Zhang XD, et al. Dual Processing of FAT1 Cadherin Protein by Human Melanoma Cells Generates Distinct Protein Products. *J. Biol. Chem.* 2011;286:28181–28191

REF3: Hou, R. & Sibinga, N. E. S. Atrophin Proteins Interact with the Fat1 Cadherin and Regulate Migration and Orientation in Vascular Smooth Muscle Cells. *J. Biol. Chem.* **284**, 6955–6965 (2009).

The additional information and the new figure 6 is much appreciated. Besides defective junctional complexes the failure to assemble a monolayer could also be part of the phenotype.

We agree with the reviewer's suggestion that the failure to assemble a monolayer could also be part of the phenotype, also it could be a secondary defect resulting from failure to form junctions with the neighbouring cells. This point has now been included in the Results section of the manuscript (page 16).

I want to thank the authors for explaining the trans-epithelial resistance measurement and the reason for the display of the data. This seems plausible.

The OCT images are very much appreciated. Could maybe also a normal OCT image be added? Could perhaps be pointed out, whether the retinal differentiation in the neighbouring region was affected? In b the retinal layering seems to be affected. I think that additional “focal planes” would be helpful.

We have now also added a normal OCT image for comparison (**Supplementary Figure 1C**). Unfortunately, we were not able to obtain additional focal planes for this patient.

Reviewer # 2 (remarks to the Author)

My only major problem with this manuscript is that it almost completely ignores what seems to me to be the most obvious and interesting function on FAT1 in this context – both molecularly and clinically. FAT1 inhibits YAP1 via phosphorylation (for review see PMID:23076869) and function loss of FAT1 has been shown to cause activation of YAP1 in a very nice recent Nature Communications paper (Nat Commun. 2018 Jul 9;9(1):2372. doi: 10.1038/s41467-018-04590-1). Given the remarkable similarity in the resulting clinical phenotypes I would suggest that looking for loss of YAP1 phosphorylation in the animal models would be a critical experiment for this paper.

We thank the reviewer for raising an interesting point. Indeed, in cell culture studies FAT1 has been shown to phosphorylate YAP1 resulting in nuclear accumulation of YAP1 (**REF1**). Following the reviewer's suggestion, we therefore compared the localization pattern and phosphorylation status of YAP1 in FAT1 shRNA treated RPE cells, and in MEF cells and mouse optic cup cryo-sections of knockout and WT mice.

We performed immunostaining of YAP/TAZ in scrambled and FAT1 shRNA treated RPE cells, but did not observe changes in the YAP/TAZ expression levels and localization pattern (**Supplementary fig.11a**). We also performed Western blot analysis to look at total YAP/TAZ and phosphorylated YAP (Ser127) levels but did not observe any significant differences between scrambled and FAT1 shRNA treated RPE cells (**Supplementary fig.11b**). Similarly, YAP/TAZ expression and localization pattern did not differ between embryonic fibroblasts (MEF) from WT and *Fat1*^{-/-} mouse embryos (**Supplementary fig.11c**). Furthermore, while in WT mouse we observed a YAP/TAZ expression pattern in the mouse optic cup similar to what has been reported previously (**REF2**), no significant changes in YAP/TAZ expression pattern were observed between mouse optic cup cryo-sections from WT and *Fat1*^{-/-} mouse (E11.5-12.5, **Supplementary fig.11d**).

While our findings appear to differ from those of the recent Nature Communications of Martin et al. (**REF1**), it is important to note that while this study and other previous studies conducted experiments in a transformed cell line (HEK293, HUVEC.), we conducted all our experiments in primary mouse and human cells. Yet, we acknowledge that additional studies on mechanisms by which FAT1 may impact YAP are warranted. Since we observed *FAT1* RNA labelling in POM it is possible that FAT1 could be involved in regulating YAP levels in the POM but not in the RPE. It is also possible that FAT1 regulates YAP localization through non-canonical pathways as is done by FAT4 (**REF3**).

We have added these new experiments to the manuscript. Please refer to page 22-23 and **Supplementary Figure 11**. Furthermore, in this regard we have mentioned the necessity for additional studies in the future.

REF1: Martin D, Degese MS, Vitale-Cross L, et al. Assembly and activation of the Hippo signalome by FAT1 tumor suppressor. Nat. Commun. 2018;9:2372.

REF2: Williamson KA, Rainger J, Floyd JA, Ansari M, Meynert A, Aldridge KV, Rainger JK, Anderson CA, Moore AT, Hurler ME, Clarke A, van Heyningen V, Verloes A, Taylor MS, Wilkie AO; UK10K Consortium, Fitzpatrick DR. Heterozygous loss-of-function mutations in YAP1 cause both isolated and syndromic optic fissure closure defects. *Am J Hum Genet.* 2014 Feb 6;94(2):295-302.

REF3: Ragni C V., Diguët N, Le Garrec J-F, et al. Amotl1 mediates sequestration of the Hippo effector Yap1 downstream of Fat4 to restrict heart growth. *Nat. Commun.* 2017;8:14582.

Abstract:

- Truncating is misleading and should be replaced with “null” or “likely null”

We changed ‘truncating’ to ‘homozygous frameshift’ throughout the manuscript.

- Suggest ptosis, which is widely used in English, as a replacement throughout the paper for blepharoptosis (which is a correct but slightly arcane term)

We replaced ‘blepharoptosis’ by ‘ptosis’ throughout the manuscript.

- “developmental ophthalmic defect” is redundant in the

We removed “developmental ophthalmic defect” from the abstract.

- Suggest replacing “our patients” with “affected individuals” as possessive terms are inappropriate in clinical reports

We replaced ‘our patients’ by ‘affected individuals’ throughout the manuscript.

- Suggest replacing “placing it exquisitely at the precise cell type and location for” with “consistent with a role in”

We replaced ‘placing it exquisitely at the precise cell type and location for’ by ‘consistent with a role in’ in the abstract.

Introduction: It is probably more accurate to say that the eye develops from the eye field within the neural plate rather than from the neural tube

We changed the Introduction according to the reviewers’ suggestion.

Methods: The different sequencing protocols that are used for each family would be best presented as a table which also includes the mean coverage and numbers of variants called across the samples sequenced by each pipeline.

We added **Supplementary Table 1** to the manuscript that summarizes the different sequencing protocols and mean depth of coverage for each family.

Reviewer #3 (Remarks to the Author):

My points were well addressed by the authors

Reviewer #1 (Remarks to the Author):

Referee comments

Referee comments on Review of NCOMMS-18-06101B. ^[SEP]In this manuscript entitled “Truncating mutations in the atypical protocadherin FAT1 cause a recessive clinical syndrome characterized by colobomatous microphthalmia, blepharoptosis, nephropathy and syndactyly” Lahrouchi, Ratbi, George and colleagues present their second revised work of a previous version of this manuscript.

I want to thank the authors. With only a few exceptions the comments to my concerns were well addressed. I especially appreciate the inclusion of a potential defective morphogenesis in the discussion.

Open issues:

Here (below), I do not get the answer the authors provide. The yellow signal is supposed to show a colocalisation. When green membranes can be found in so many cells, it could well be that the hypothesis of a colocalisation of FAT1 especially with tight junctions is not proper.

“Changing the image in 4b is not solving the question/ problem from before. What was the reason for the green membranes?”

2/6

The F-Actin and ZO-1 staining (which are green) tend to be more specific and localized to the apical tight junctions only, as such they have a better signal/noise ratio. The FAT1 antibody is localized at the cell membrane and cytoplasm and as a result the signal/noise ratio with this antibody is not optimal.”

Here (below), I think that the authors do not provide a sufficient answer with respect to the variability of the different phenotype intensities, resulting from the morpholino induced knockdown.

In figure 3 (old version c) a different phenotype was presented, not well fitting to the sections provided in f,g. The new pictures fit much better. However, the variability of the phenotype should be addressed. It was mentioned that morpholino based knock down can result in mild, moderate and severe phenotypes. Would the authors still claim that in all distinct categories the same pathogenesis is the basis? I appreciate that the other images are added to the supplemental figures.

The variability of the morpholino phenotype is addressed in Supplementary Figure 4. Since it is well known that morpholino mediated knockdown results in a variable phenotype (ranging from mild to severe) we used 2 different morpholino concentrations and observed a mild phenotype at low concentration and severe defects were noted at high morpholino concentration. Whether this variability is due to a morpholino dosage effect or other effects is currently unknown. Notwithstanding, the observations we made in the Fat1 morpholino knock down zebrafish were confirmed in the other two models that we studied, namely CRISPR FAT1 mutant zebrafish and in Fat1^{-/-} mouse embryos.

As a minor point: It would be helpful for the reader if the OCT images could be annotated

Reviewer #2 (Remarks to the Author):

The authors have addressed all of the points in my original review and I am grateful for their considered and positive response

RESPONSE TO REVIEWERS' COMMENTS:

Reviewer #1 (Remarks to the Author):

Referee comments

Referee comments on Review of NCOMMS-18-06101B. In this manuscript entitled “Truncating mutations in the atypical protocadherin FAT1 cause a recessive clinical syndrome characterized by colobomatous microphthalmia, blepharoptosis, nephropathy and syndactyly” Lahrouchi, Ratbi, George and colleagues present their second revised work of a previous version of this manuscript.

I want to thank the authors. With only a few exceptions the comments to my concerns were well addressed. I especially appreciate the inclusion of a potential defective morphogenesis in the discussion.

Open issues:

Here (below), I do not get the answer the authors provide. The yellow signal is supposed to show a colocalisation. When green membranes can be found in so many cells, it could well be that the hypothesis of a colocalisation of FAT1 especially with tight junctions is not proper.

We agree with the reviewer that the current data is not enough to support this claim and more experimentation would be required in this regard. As such we have decided to drop the figure 4 for the manuscript and made corresponding changes as suggested by the reviewer and the editor.

“Changing the image in 4b is not solving the question/ problem from before. What was the reason for the green membranes? 2/6 The F-Actin and ZO-1 staining (which are green) tend to be more specific and localized to the apical tight junctions only, as such they have a better signal/noise ratio. The FAT1 antibody is localized at the cell membrane and cytoplasm and as a result the signal/noise ratio with this antibody is not optimal. “

Here (below), I think that the authors do not provide a sufficient answer with respect to the variability of the different phenotype intensities, resulting from the morpholino induced knockdown.

In figure 3 (old version c) a different phenotype was presented, not well fitting to the sections provided in f,g. The new pictures fit much better. However, the variability of the phenotype should be addressed. It was mentioned that morpholino based knock down can result in mild, moderate and severe phenotypes. Would the authors still claim that in all distinct categories the same pathogenesis is the basis? I appreciate that the other images are added to the supplemental figures.

The variability of the morpholino phenotype is addressed in Supplementary Figure 4. Since it is well known that morpholino mediated knockdown results in a variable phenotype (ranging from mild to severe) we used 2 different morpholino concentrations and observed a mild phenotype at low concentration and severe defects were noted at high morpholino concentration. Whether this variability is due to a morpholino dosage effect or other effects is

currently unknown. Notwithstanding, the observations we made in the Fat1 morpholino knock down zebrafish were confirmed in the other two models that we studied, namely CRISPR FAT1 mutant zebrafish and in Fat1^{-/-} mouse embryos.

We can currently not provide the definite answer for the variability observed in phenotype. Since morpholino would be able to block the translation of maternally deposited transcripts they are usually expected to exhibit a severe phenotype. However, the mutants (derived from heterozygous cross) will still retain some levels of WT transcripts that are maternally deposited. As such we the variability would be a compounding effect of injection efficiency, injection volume and diffusion efficiency of the morpholino in the developing morphant embryo, which we think might contribute towards the variable phenotype observed with the morpholino knockdown experiment. As suggested by the editor we have decided to move the morpholino data to the supplementary Information of the manuscript.

As a minor point: It would be helpful for the reader if the OCT images could be annotated

We have added an annotation the OCT images.

Reviewer #2 (Remarks to the Author):

The authors have addressed all of the points in my original review and I am grateful for their considered and positive response.

We thank the reviewer for his/her critical review.